# *KCNA2* variants cause dilated cardiomyopathy, obesity and sleep apnea through RAC-ERK pathway

Prasanth Chimata[1,2], Sahil Lall[3], Tarmo Annilo[4], M K Mathew[3], Andres Metspalu[4], Jayaprakash Shenthar[5] & Perundurai S Dhandapany 🔟 [1✉]

## Abstract

Dilated cardiomyopathy (DCM) is a heart condition characterized by systolic and diastolic dysfunction. In many instances, patients with DCM coexist with obesity and sleep apnea. It is unclear whether genetic variants contribute to the combined phenotypes of DCM, obesity, and sleep apnea. Here, using next-generation sequencing, we identified pathogenic *KCNA2* variants in patients of diverse ancestry with DCM, Obesity, and Sleep Apnea (termed DOSA). Electrophysiological and biochemical assays using biosensors revealed loss of membrane current due to trafficking defects in cells expressing *KCNA2* variants. Furthermore, cellular models including patient-specific iPSC cardiomyocytes and organoid models displayed RAC1-ERK1/2 hyperactivation in disease pathogenesis. A *Drosophila* model expressing KCNA2 variant showed DOSA-like phenotypes which was rescued using RAC1 inhibitors. Our results provide the first evidence that *KCNA2* variants can lead to DOSA phenotypes, further expanding the genetic regulatory roles of potassium channels in human diseases.

**Keywords** Dilated Cardiomyopathy; Obesity; Sleep Apnea
**Subject Categories** Cardiovascular System; Genetics, Gene Therapy & Genetic Disease; Metabolism

## Introduction

*KCNA2* encodes a potassium voltage-gated channel belonging to the Kv1 subfamily (Imbrici et al, 2021; Syrbe et al, 2015). It is a delayed rectifier voltage-gated channel that is vital for the repolarization phase of the cardiac action potential (Syrbe et al, 2015; Long et al, 2017). These processes activate various downstream signaling, including RAC1-ERK1/2 (Yang et al, 2020; Clerk et al, 2001). To date, *KCNA2* pathogenic variants have been reported in epileptic encephalopathies (Syrbe et al, 2015). However, its shared genetic roles in DCM, obesity, and sleep apnea are not yet understood. Notably, DCM coexists with obesity (11–14%) (Ebong et al, 2014) and sleep apnea (40–60%) (Oldenburg et al, 2007; Yeghiazarians et al, 2021). Here, we describe *KCNA2* pathogenic variants in patients with DCM, obesity, and sleep apnea (DOSA). The DOSA-related *KCNA2* variants hyperactivated RAC1-ERK1/2 signaling in 2D/3D patient-derived iPSC cardiomyocytes/organoids and transgenic *Drosophila* models, which could be reversed by RAC1 inhibitors.

## Results and discussion

### Identification of *KCNA2* as a gene for DOSA phenotypes

To identify whether pathogenic variants cause combined phenotypes of DCM, obesity, and sleep apnea, we conducted exome and targeted sanger sequencing on an affected South Asian Indian (SAI) family (P2-P3) (Fig. 1A). This analysis revealed a novel pathogenic heterozygous *KCNA2* variant (NM_001204269, c.551 C > A) in proband P3, resulting in an amino acid substitution (p.T184K) in the protein (Fig. 1A). This variant was not present in his healthy mother (P2, familial control). In addition, we identified a homozygous KCNA2 variant (NM_001204269, c.551 C > T, p.T184I) in an independent cohort of Indian DCM patients (P8, 1 out of 436 alleles) (Dhandapany et al, 2014) (Fig. 1B,C). The clinical registry indicated that P8 also exhibited obesity and sleep apnea. Neither of the variants was observed in region-specific SAI controls (0 of 3442 alleles) nor in global populations with mixed ancestry (0 of 1,614,324 alleles) (Fig. 1C). Both P3 and P8 tested negative for gene variants known to cause DCM, sleep apnea, and obesity (Appendix Fig. S1). Subsequently, we identified another *KCNA2* variant (NM_001204269: c.565 C > T, p.R189W) in Estonian biobank samples, present in fewer than 5 individuals out of 2956 alleles. This variant was absent in the ancestry-matched Estonian controls (0 of 6594 alleles) but exhibited an extremely low MAF of 0.000004 (7 of 1,614,032 alleles) in other global populations. The *KCNA2* variants altered the highly conserved threonine (p.T184) and arginine (p.R189) residues on the S1–S2 loop (Fig. 1D,E). Various biocomputational tools predicted these variants to be pathogenic (Fig. 1F).

[1]Cardiovascular Development and Disease Mechanisms, Institute for Stem Cell Science and Regenerative Medicine (BRIC-inStem), Bengaluru, Karnataka 560065, India. [2]Manipal Academy of Higher Education, Manipal, Karnataka 576104, India. [3]National Centre for Biological Sciences, Tata Institute of Fundamental Research, Bengaluru, Karnataka 560065, India. [4]Estonian Genome Centre, Institute of Genomics, University of Tartu, Tartu 51010, Estonia. [5]Sri Jayadeva Institute of Cardiovascular Sciences and Research, Jayanagar, Bengaluru 560069 Karnataka, India. ✉E-mail: dhan@instem.res.in

## KCNA2 variants display loss of whole-cell current due to membrane trafficking

To assess whether the KCNA2 variants influence current conduction or channel trafficking, we introduced individual representative KCNA2 constructs (wild-type, p.T184K, p.R189 W, and p.T184I) into the cells, along with a reporter (EGFP) and surface protein marker (HA). Electrophysiological and immunostaining analyses of cells expressing the KCNA2 variants revealed a loss of detectable whole-cell current. This correlated with a failure of membrane localization of the variant proteins compared to the wild-type (Fig. 2; Appendix Fig. S2). These results suggest that KCNA2 variants result in defects in conduction and in cell surface trafficking.

## KCNA2 variants increase cell size, fetal gene markers, and RAC1-ERK1/2

To determine whether KCNA2 variants can independently trigger a heart failure phenotype, we expressed the variant constructs in cardiomyocytes alongside the wild-type control to study their effect (Appendix Figs. S3–S7). Immunofluorescence analysis showed that the KCNA2 variant proteins caused a significant increase in cell size compared to cardiomyocytes expressing WT (Appendix Figs. S3 and S7A,B). Subsequently, we assessed the mRNA levels of several known heart failure markers, such as atrial natriuretic peptide (*Nppa*), brain natriuretic peptide (*Nppb*), skeletal muscle alpha-actin (*Acta1*), and myosin heavy chain 7 (*Myh7*), in cardiomyocytes expressing KCNA2 WT and variants, respectively. We observed a marked increase in heart failure markers in cardiomyocytes expressing KCNA2 variants compared to those in the control cardiomyocytes (Appendix Figs. S3C and S7C). These results suggest that KCNA2 variants induce heart failure.

Next, we examined the downstream signaling pathways modulated by the KCNA2 variants using immunoblots of lysates expressing WT and variant proteins (Appendix Figs. S3D–G and S7D–I). We focused on RAC1 signaling because of its involvement in membrane repolarization and subsequent endocytic trafficking of potassium channels (Yang et al, 2020; Boyer et al, 2009). In addition, RAC1 is known to activate downstream ERK1/2 in cardiac hypertrophy (Clerk et al, 2001; Brown et al, 2006). We observed hyperactivation of RAC1-ERK1/2 signaling in KCNA2 variant-expressing cardiomyocytes compared to WT (Appendix Figs. S4B–D and S7D–H). These effects were rescued upon treatment with RAC1 inhibitors (RACi) (EHT1864 or Simvastatin) (Shutes et al, 2007; Chen et al, 2008; Takemoto et al, 2001) (Appendix Figs. S4, S5, S6, and S7). These findings suggest that RAC1-ERK1/2 signaling plays a vital role in KCNA2-induced heart failure phenotypes.

## *KCNA2* patient-specific iPSC-derived cardiomyocytes and organoids exhibit heart failure

Furthermore, we generated a patient-specific iPSC line with KCNA2 p.T184K (P3) along with a healthy familial control iPSC (P2, mother of P3) (Appendix Figs. S8 and S9) and differentiated them into cardiomyocytes (iPSC-CMs). The patient iPSC-CMs exhibited significantly increased sarcomeric disorganization (80% of the total CMs), fetal gene expression (Fig. 3A,B). In addition, 3D cardiac organoids generated from KCNA2 p.T184K iPSCs also displayed increased sarcomeric disorganization, fetal gene expression, and hyperactivation of RAC1-ERK1/2 signaling compared to the familial healthy control iPSC-cardiac organoids (Appendix Fig. S10).

In addition, we performed a transcriptomic analysis of untreated and simvastatin-treated cardiomyocytes derived from KCNA2 p.T184K iPSCs. The gene ontology (GO) molecular function of the transcripts revealed significantly differentially expressed genes related to hypertrophy, cellular trafficking, and lipid-related pathways (Fig. 3C–E; Appendix Fig. S11). Further, we observed hyperactivation of RAC1-ERK1/2 signaling compared to the familial healthy control iPSC-CMs. Simvastatin treatment rescued these parameters along with RAC1-ERK1/2 signaling (Fig. 3D–G), underscoring the importance of the involvement of RAC1 in the pathogenesis.

## *Drosophila* model of *KCNA2* displayed DOSA-like phenotypes

To assess the systemic impact of the *KCNA2* variant, we utilized a representative *Drosophila* model (Cirelli et al, 2005) that mimics the patient variants at the p.Thr 184th position, which is referred to here as KI. Phalloidin staining of F-actin in the hearts of KI flies revealed disorganized myofibrils with abnormal gaps between the A2-A3 segments of the myofibers (Fig. 4A). In addition, semi-intact adult KI fly heart tubes exhibited abnormal cardiac contractility and reduced contraction amplitude (Fig. 4B). These data suggest cardiac dysfunction in KI flies. Thus, our study unravels a previously unrecognized pathophysiological role for KCNA2 in the heart.

To investigate the effects of obesity in KI flies, we performed Oil Red O and Nile Red staining, which showed increased abdominal fat body mass and lipid droplets in KI compared to wild-type flies (Fig. 4C,D). Consistently, the total triglyceride (TGA) levels were significantly increased in the KI flies (Fig. 4E). These data suggest that KI flies recapitulate the obesity phenotypes. The KI flies exhibited reduced sleep phenotypes (Fig. 4F). Thus, at the systemic level, KI flies displayed DOSA-like phenotypes.

Next, we treated the KI flies with simvastatin, a RAC1 inhibitor, to understand the rescue of DOSA phenotypes. Our findings revealed that simvastatin effectively reversed myofibrillar disarray and improved cardiac contractility in the KI flies (Appendix Fig. S12A–C). The treatment also reduced TGA levels and lipid droplet size in fat bodies (Appendix Fig. S12D,E). Also, treatment of KI flies with simvastatin demonstrated a marked improvement in overall sleep and its episodes by modulating the number and duration of sleep bouts compared with the untreated group (Appendix Fig. S12F–H). Taken together, our findings provide the first evidence that KCNA2 variants cause DOSA by activating the RAC1-ERK pathway, which can be rescued by RACi.

We report *KCNA2* variants as a new cause for combined patient phenotypes involving DCM, obesity, and sleep apnea (DOSA). Our genetic screening identified KCNA2 p.T184K and p.T184I as novel gene variants associated with DOSA. These variants were absent in public human population genome reference datasets of various ethnicities. We replicated our observations in an international patient cohort with similar phenotypes comprising Estonian biobank samples by identifying another *KCNA2* variant, p.R189W, respectively. This variant was absent in Estonian controls ($n = 3297$). However, it was present at an ultra-rare frequency in African American and European (non-Finnish) populations. The odds ratio for the *KCNA2* p.R189W variant in Estonian Biobank

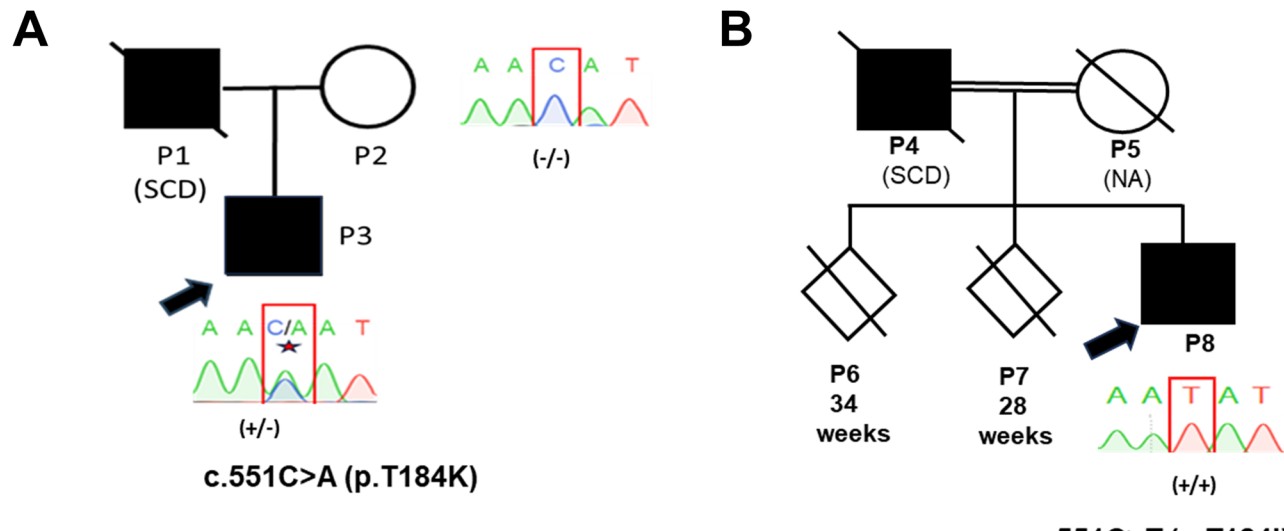

**C**

| Cohort | Amino Acid (AA) | AF (gnomAD) | Age/Sex | LVIDd (mm) | LVEF% | Mitral regurgitation | Sleep Apnea | Obesity |
|--------|-----------------|-------------|---------|------------|-------|----------------------|-------------|---------|
| SAI | p.T184K | Novel | 34y/M | 55 | 30 | Mild | Yes | Yes |
| SAI | p.T184I | Novel | 17y/M | 70 | 18 | Moderate | Yes | Yes |

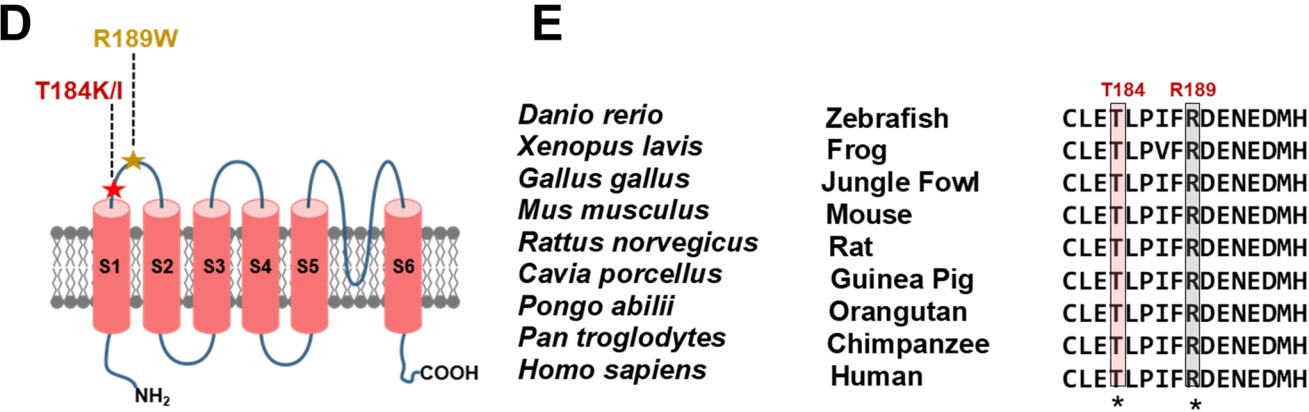

**F**

| Exon | Nucleotide change | Amino acid change | SIFT | Polyphen2 | PROVEAN | PANTHER-PSEP | CADD | Mutation-taster | REVEL |
|------|-------------------|-------------------|------|-----------|---------|--------------|------|------------------|-------|
| 3 | c.551C>A | p.T184K | D (0) | D (1) | D(-5.7) | PD(0.85) | DC(26.7) | D(1) | 0.8 |
| 3 | c.551C>T | p.T184I | D (0) | D (1) | D(-5.7) | PD(0.85) | DC(27.8) | D(1) | 0.818 |
| 3 | c.565C>T | p.R189W | D(0) | D (0.997) | D(-6.6) | PD(0.74) | DC(29.4) | D(1) | 0.541 |

**Figure 1.  Molecular genetics of KCNA2 variants.**

(A, B) Pedigrees of SAI patients. Filled symbols represent affected individuals. SCD sudden cardiac death. The (+) and (−) signs indicate the presence and absence of amino acid changes. NA not available. (C) Allele frequencies (AF) of KCNA2 variants. LVIDd left ventricular internal dimension end diastole, LVEF left ventricular ejection fraction. (D) The locations of the KCNA2 residues were altered. (E) Protein alignment of KCNA2 across species, highlighting altered AA in patients. (F) In silico analysis of KCNA2 variants (D damaging, DC disease-causing, PD probably damaging).

cases (1/1478) was 78.0 compared to all other global populations, including African American and European (non-Finnish) in gnomAD (7/807016) (95% CI 9.59 to 634.408; $P < 0.0001$). These data support a strong genetic association for KCNA2 in DOSA.

All the identified variants were located in the S1–S2 loop of KCNA2 in DOSA. In contrast, other disease-causing variants, such as those in epilepsy, are situated predominantly outside the S1–S2 loop (Syrbe et al, 2015). Computational tools have predicted that these variants cause considerable damage to protein structure and function. In addition, we demonstrated a diminished whole-cell current and impaired cell surface trafficking for these variants in cellular models. This is in line with previous research, where this region was shown to be involved in conduction and cell surface localization (Mckeown et al, 2008). Similarly, KCNA2 variants associated with epilepsy also affect cell surface localization and cause conduction defects (Syrbe et al, 2015; Nilsson et al, 2022). In line with this, we observed similar effects in KCNA2 variants associated with DOSA. However, we did not observe epilepsy in patients with DOSA, likely due to positional effects. Unlike epilepsy associated variants, which are located in the S2, S3, S4, and S6 transmembrane segments, the DOSA-related variants are located in the S1–S2 loop (Syrbe et al, 2015; Nilsson et al, 2022).

Ion channels are known to regulate the plasma membrane potential ($V_m$) due to altered ion flux, leading to RAC1 activation (Yang et al, 2020). In this study, we showed that KCNA2 variants hyperactivate RAC1-ERK1/2 signaling. Transgenic mice with constitutively active RAC1 have been shown to induce heart failure phenotypes (Reil et al, 2010; Sussman et al, 2000) and activate ERK1/2 (Clerk et al, 2001). Also, RAC1 activity is known to be upregulated in patients with heart failure (Maack et al, 2003). Consistent with these data, we demonstrated that KCNA2 variants induced hypertrophy in cultured cardiomyocytes by increasing cell size, reactivating the fetal gene program, and activating the RAC1-ERK1/2 signaling pathways.

Further, KCNA2-positive patient 2D and 3D models, including iPSC-CMs and cardiac organoids, recapitulated heart failure phenotypes, including sarcomeric disarray, fetal gene markers, and activation of the RAC1-ERK1/2 pathway. Transcriptome analysis revealed the dysregulation of genes and pathways associated with endocytic trafficking, conduction, calcium handling, and cardiac maturation in the KCNA2 patient cardiomyocytes compared to those in healthy familial controls. Treatment of the KCNA2 patient iPSC-CMs with simvastatin rescued the sarcomeric disarray and normalized RAC1-ERK1/2 activity, similar to the controls. Notably, simvastatin inhibits the prenylation of RAC1, thereby preventing its membrane localization and its activity (Chen et al, 2008; Takemoto et al, 2001). Further, the transcriptomic analysis of the treated KCNA2 iPSC-CMs also showed normalization of heart failure, endocytic trafficking, and lipid-related mRNA levels compared to the controls. These data support the notion that the RAC1-ERK1/2 signaling pathway is crucial for inducing DCM in mutant iPSC-CMs.

The KI Drosophila model exhibited abnormal cardiac contractility, myofibrillar organization, lipid accumulation, and sleep disorders that were rescued upon treatment with simvastatin. Inhibition of RAC1 through simvastatin is known to improve cardiac function in aging Drosophila and reduce triglyceride levels in human subjects (Spindler et al, 2012; Isley et al, 2006). RAC1 is known to regulate adipogenesis in mouse models, as demonstrated by a study showing that the white adipose tissue in RAC1 knockout mice attenuated the formation of lipid droplets, indicating that RAC1 mediates lipid metabolism (Hasegawa et al, 2023). Active RAC1 mutations in humans have been reported to be involved in cerebral abnormalities, including sleep disturbances (Banka et al, 2022; Reijnders et al, 2017). Thus, RAC1 signaling is important for KCNA2 mutant-induced cardiac, obesity, and sleep disturbance-associated pathogenesis. Notably, KCNA2 mutants are known to affect sleep in Drosophila models (Cirelli et al, 2005). Based on these findings, we observed that KCNA2 genotype-positive patients also suffer from sleep apnea. Interestingly, the recessive KI flies replicated the autosomal dominant DOSA-like phenotypes of the patients. While certain human autosomal dominant diseases have been observed to exhibit such a recessive pattern in Drosophila, this phenomenon has nonetheless advanced the understanding of disease pathogenesis (Gross et al, 2020; The et al, 1997; Yokoi et al, 2016; Clements et al, 2009).

Taken together, we provide the first evidence that clinically relevant pathological KCNA2 variants contribute to the DOSA phenotypes targeting RAC1-ERK1/2 signaling that can be rescued by RAC1 inhibitors.

## Limitations

Our research primarily aims to determine the pathological roles of KCNA2 by utilizing Semi-intact Drosophila heart models to examine cardiac abnormalities. Further investigations employing Optical Coherence Tomography (OCT) will provide more insights into cardiac structure and function. In addition, we have demonstrated evidence for RAC1-ERK1/2 dependent signaling in KCNA2-induced pathogenesis; however, the potential involvement of other pathways or mechanisms cannot be ruled out.

## Methods

**Reagents and tools table**

| Reagent/resource | Reference or source | Identifier or catalog number |
|---|---|---|
| **Experimental models** | | |
| H9c2 | ATCC | CRL-1446 ™ |
| HEK293T | ATCC | CRL-3216 ™ |
| COS-7 | ATCC | CRL-1651 ™ |

| Reagent/resource | Reference or source | Identifier or catalog number |
|---|---|---|
| Human Ipsc control | This Study | NA |
| Human Ipsc Patient | This Study | NA |
| Drosophila Sh[mns] | Bloomington Drosophila Stock Center | 24149 |
| Drosophila Canton S | Bloomington Drosophila Stock Center | 64349 |
| **Recombinant DNA** | | |
| pXOOM-KCNA2 constructs | This study | NA |
| pEGFP-C1 N-terminal fused rat Kv1.2 mutant constructs with an extracellular HA site (between transmembrane helices S1 and S2) | This study | NA |
| **Antibodies** | | |
| Anti-HA conjugated with Alexa Fluor 647 | BioLegend | 682404 |
| Anti-extracellular signal-regulated kinase 1/2 (ERK1/2) | Cell Signalling Technologies | 9102S |
| Anti-phospho-ERK1/2(Thr202/Tyr204) | Cell Signalling Technologies | 9101S |
| Anti-glyceraldehyde-3-phosphate dehydrogenase (GAPDH) | Thermo Fisher Scientific | MA5-15738 |
| Anti RAC1 | Cell Signalling Technologies | From the rac1 activity Kit (8815) |
| Anti KCNA2 (Kv1.2) | Abcam | ab192758 |
| Anti-mouse HRP-conjugated secondary | Thermo Fisher Scientific | 31430 |
| Anti-rabbit HRP-conjugated secondary | Thermo Fisher Scientific | 31460 |
| Anti-α-sarcomeric actinin | Thermo Fisher Scientific | 701914 |
| Anti-Nanog | Santa Cruz | sc-293121 |
| Anti-TRA-1-81 | Santa Cruz | sc-21706 |
| Anti-SSEA4 | Santa Cruz | sc-59368 |
| Goat anti-Rabbit IgG (H+L) Cross-Adsorbed Secondary Antibody, Alexa Fluor™ 488 | Thermo Fisher Scientific | A-11008 |
| Goat anti-Mouse IgG (H+L) Cross-Adsorbed Secondary Antibody, Alexa Fluor™ 546 | Thermo Fisher Scientific | A-11003 |
| **Oligonucleotides and other sequence-based reagents** | | |
| PAX6 forward | Eurofins | AACGATAACATACCAAGCGTGT |
| PAX6 reverse | Eurofins | GGTCTGCCCGTTCAACATC |
| SOX1 forward | Eurofins | GAGTGGAAGGTCATGTCCGAGG |
| SOX1 reverse | Eurofins | CCTTCTTGAGCAGCGTCTTGGT |
| HAND1 forward | Eurofins | AACTCAAGAAGGCGGATGG |
| HAND1 reverse | Eurofins | GGAGGAAAACCTTCGTGCT |
| TBXT forward | Eurofins | GGTCCAGCCTTGGAATGCCT |
| TBXT reverse | Eurofins | CCGTTGCTCACAGACCACAG |

| Reagent/resource | Reference or source | Identifier or catalog number |
|---|---|---|
| BMP4 forward | Eurofins | GCACTGGTCTTGAGTATCCTG |
| BMP4 reverse | Eurofins | TGCTGAGGTTAAAGAGGAAACG |
| SOX17 forward | Eurofins | GTGGACCGCACGGAATTTGA |
| SOX17 reverse | Eurofins | GCTGTCGGGGAGATTCACAC |
| FOXA2 forward | Eurofins | ATGCACTCGGCTTCCAGTATG |
| FOXA2 reverse | Eurofins | TGTTCATGCCGTTCATCCCC |
| RNU6-1 forward | Eurofins | ATTGGAACGATACAGAGAAGATTAG |
| RNU6-1 reverse | Eurofins | AATATGGAACGCTTCACGAAT |
| Human ACTA1 forward | Eurofins | TCTCACCGACTACCTGATGAA |
| Human ACTA1 reverse | Eurofins | AGCACAGCTTCTCCTTGATG |
| Human MYH6 forward | Eurofins | ATATACCTACTCGGGCCTCTT |
| Human MYH6 reverse | Eurofins | GTCGGAGATGGAGAAGATGTG |
| Human NPPA forward | Eurofins | TTGCTGGACCATTTGGAAGA |
| Human NPPA reverse | Eurofins | GCTTCTTCATTCGGCTCACT |
| Human NPPB forward | Eurofins | TCCTGCTCTTCTTGCATCTG |
| Human NPPB reverse | Eurofins | GTAACCCGGACGTTTCCAA |
| Human 18s rRNA forward | Eurofins | GTAACCCGTTGAACCCCATT |
| Human 18s rRNA reverse | Eurofins | CCATCCAATCGGTAGTAGCG |
| Human MYH7 forward | Eurofins | TGAAGGAGGACCAGGTGAT |
| Human MYH7 reverse | Eurofins | GTAGCGATCCTTGAGGTTGTAG |
| Rattus novergicus Myh7 forward | Eurofins | CCTCGCAATATCAAGGGAAA |
| Rattus novergicus Myh7 reverse | Eurofins | TACAGGTGCATCAGCTCCAG |
| Rattus novergicus Acta1 forward | Eurofins | CCTGGACTTCGAGAATGAGATG |
| Rattus novergicus Acta1 reverse | Eurofins | CGATAAAGGAAGGCTGGAAGAG |
| Rattus novergicus Nppb forward | Eurofins | AGATGATTCTGCTCCTGCTTT |
| Rattus novergicus Nppb reverse | Eurofins | ATCGTGGATTGTTCTGGAGAC |
| Rattus novergicus 18s rRNA forward | Eurofins | GGAACTGAGGCCATGATTAAGA |
| Rattus novergicus 18s rRNA reverse | Eurofins | CAAATGCTTTCGCTCTGGTTC |
| Rattus novergicus Nppa forward | Eurofins | ATTTCAAGAACCTGCTAGACC |
| Rattus novergicus Nppa reverse | Eurofins | TTTTCAAGAGGGCAGATCTAT |
| KCNA2 forward | Eurofins | TCGGTTTTATGAGCTGGGAGAAG |
| KCNA2 reverse | Eurofins | ACCTAGAATCTGGAGACCTTTGG |
| Sev forward (from the kit) | Eurofins | GGATCACTAGGTGATATCGAGC |
| Sev reverse (from the kit) | Eurofins | ACCAGACAAGAGTTTAAGAGATATGTATC |
| KOS transgene forward (from the kit) | Eurofins | ATGCACCGCTACGACGTGAGCGC |
| KOS transgene reverse (from the kit) | Eurofins | ACCTTGACAATCCTGATGTGG |
| GAPDH forward | Eurofins | GAAGGTGAAGGTCGGAGTC |
| GAPDH reverse | Eurofins | GAAGATGGTGATGGGATTTC |
| **Chemicals, enzymes, and other reagents** | | |
| Dulbecco's modified Eagle's medium | Gibco | 12800017 |
| Fetal bovine serum | Gibco | A5256701 |

| Reagent/resource | Reference or source | Identifier or catalog number |
|---|---|---|
| penicillin/streptomycin | Gibco | 15140122 |
| jetPRIME | Sartorius | 101000001 |
| EHT1864 | Cayman Chemical | 17258 |
| Simvastatin | Cayman Chemical | 10010344 |
| 35 mm glass-bottom dish | Ibidi | 81218-200 |
| Hoechst | Thermo Fisher Scientific | 62249 |
| Trypsin | Thermo Fisher Scientific | 25200056 |
| Zombie Violet | BioLegend | 423113 |
| RIPA Buffer | Sigma-Aldrich | R0278 |
| PMSF | Thermo Fisher Scientific | 36978 |
| Protease and phosphatase inhibitor cocktail | Thermo Fisher Scientific | 78440 |
| Polyvinylidene fluoride (PVDF) membrane | BioRad | 1620177 |
| WESTAR ANTARES chemiluminescence reagent | Cyanagen | XLS142,0250 |
| Bovine serum albumin | HiMedia | MB083 |
| Phalloidin Alexa Fluor 546 | Thermo Fisher Scientific | A22283 |
| TRIzol | Thermo Fisher Scientific | 423113 |
| Verso cDNA Synthesis Kit | Thermo Fisher Scientific | AB1453A |
| PowerUp™ SYBR™ Green Master Mix | Thermo Fisher Scientific | A25742 |
| Active RAC1 Detection Kit | Cell Signalling Technologies | 8815 |
| CytoTune-iPS 2.0 Sendai virus-based reprogramming kit | Thermo Fisher Scientific | A16517 |
| StemPro™ -34 | Thermo Fisher Scientific | 10639011 |
| SCF | Thermo Fisher Scientific | PHC2115 |
| FLT-3 | Thermo Fisher Scientific | PHC9414 |
| IL-3 | Thermo Fisher Scientific | PHC0034 |
| IL-6 | Thermo Fisher Scientific | PHC0065 |
| Vitronectin | Thermo Fisher Scientific | A14700 |
| Essential 8 | Thermo Fisher Scientific | 10639011 |

| Reagent/resource | Reference or source | Identifier or catalog number |
|---|---|---|
| StemFlex | Thermo Fisher Scientific | A3349401 |
| ReLeSR™ | Stem Cell Technologies | 100-0483 |
| RPMI | Thermo Fisher Scientific | 72400047 |
| B27 minus insulin | Thermo Fisher Scientific | A1895601 |
| CHIR99021 | Sigma-Aldrich | SML1046 |
| IWP2 | Sigma-Aldrich | I0536 |
| B27 supplement | Thermo Fisher Scientific | 17504044 |
| StemPro Accutase | Thermo Fisher Scientific | A1110501 |
| RevitaCell Supplement | Thermo Fisher Scientific | A2644501 |
| ultra-low attachment round-bottom 96-well plate | Corning, Costar | 7007 |
| BMP4 | Sigma-Aldrich | GF302 |
| Activin A | Sigma-Aldrich | SRP3003 |
| Wnt-C59 | Sigma-Aldrich | AMBH2D6FA48B |
| coconut oil | Sigma-Aldrich | C1758 |
| Triglyceride Quantification Colorimetric/Fluorometric Kit | Sigma-Aldrich | TR0100 |
| Pierce BCA protein estimation kit | Thermo Fisher Scientific | 23225 |
| Nile Red | Thermo Fisher Scientific | N1142 |
| Alexa 647-conjugated Phalloidin | Thermo Fisher Scientific | A22287 |
| Oil Red O | Sigma-Aldrich | O1391 |
| **Software** | | |
| ImageJ | | |
| PySolo | Gilestro and Cirelli, 2009 | |
| MUSCLEMOTION | Sala et al, 2018 | |
| ANNOVAR | Wang et al, 2010 | |
| Kaluza Analysis Software | Beckman Coulter | Kaluza Analysis Software, Beckman Coulter |
| **Other** | | |
| RNA seq Data | This Study | NCBI BioProject Submissions: BioProject ID PRJNA1370669 |

## Ethics declaration

The samples were obtained from Sri Jayadeva Institute of Cardiovascular Sciences and Research, Bengaluru. This study was approved by the Institutional Ethical Committees of Jayadeva Institute of Cardiovascular Sciences and Research and Institute for Stem Cell Science and Regenerative Medicine, Bangalore (Reference number: inStem/IEC-10/001). Informed consent was obtained from all subjects, and the experiments conformed to the principles set out in the WMA Declaration of Helsinki and the Department of Health and Human Services Belmont Report.

## Clinical evaluation and diagnostic criteria

Family members underwent cardiac health evaluations using echocardiography. A standard international protocol was followed in diagnosing the DCM based on our previous publication (Dhandapany et al, 2014). Individuals were diagnosed with DCM when echocardiographic assessments demonstrated diminished left ventricular systolic function, as evidenced by a left ventricular ejection fraction (LVEF) of <0.45 and/or fractional shortening <0.25. In addition, an enlarged left ventricle was indicated by a left ventricular end-diastolic dimension >117% of the expected value, adjusted for age and body surface area. This diagnosis was established in the absence of other cardiac or systemic etiologies, such as coronary artery disease and valvular disorders.

A clinically healthy family member (P2) served as an internal control to identify possible mutations responsible for the disease.

## Exome sequencing and analysis

Indian DNA samples were isolated from the patient and the family members, and whole-exome sequencing and analysis were executed using an established pipeline previously published by us (Jain et al, 2022). In summary, the exome DNA library was prepared using the SureSelect V5 Enrichment Kit (Agilent). Each exome was sequenced with 100 bp paired-end reads at 100× coverage, resulting in 6 GB of data. The capture kit's target region had an average coverage of 88, with 85% exceeding 30× and 98% surpassing 10×. Trimmomatic was used to filter out low-quality reads and adapters. The processed reads were aligned to the human reference genome (GRCh38) using the Burrows-Wheeler aligner (v0.7.17). Following this, Picard tools (v 2.9.0) were utilized to detect and remove PCR duplicates, and variant calling was executed using HaplotypeCaller from the Genome Analysis Tool Kit (GATK v4.1.9.0). ANNOVAR was used for variant annotation (Wang et al, 2010). To identify the associated variants, they were filtered for deleterious or pathogenic effects using in silico tools, such as Polymorphism Phenotyping v2 (PolyPhen2), Sorting Intolerant From Tolerant (SIFT), Protein Analysis Through Evolutionary Relationships Position-specific Evolutionary Preservation (PANTHER-PSEP), MetaDome, Combined Annotation Dependent Depletion (CADD), Mutation Taster, and Mendelian Clinically Applicable Pathogenicity (M-CAP) score. Variants were considered pathogenic or deleterious if predicted by the majority of in silico tools (4 of 6) and were evaluated against the American College of Medical Genetics criteria. The primary parameters for prioritizing variants included: (i) coding regions, (ii) ultra-rare variants with a MAF ≤ 0.01% (0.0001), and (iii) novel variants in public human population genome reference datasets

across various ethnicities. Allele frequency was estimated using reference datasets such as gnomAD (v4.1), the 1000 Genomes Project, the Korean Exome Information Database (KoEXID), Genome Asia 100 K, IndiGenomes, the Exome Aggregation Consortium (ExAC), and the in-house database of healthy aging Indian exomes (i-DHANS). Amino acid conservation across species was assessed by comparing the protein sequences of different vertebrate species using Clustal Omega online software (Madeira et al, 2024). For resequencing and confirmation of the respective pathogenic variants, Sanger sequencing was employed using KCNA2 primers (Forward: TCGGTTTTATGAGCTGGGAGAAG and Reverse: ACCTAGAATCTGGAGACCTTTGG). We followed the same methodology for other genes involved in DCM, obesity, and sleep apnea. The Estonian sequencing and analysis were detailed in a previous publication (Mitt et al, 2017).

## Patch-clamping

Cells were cultured in high-glucose DMEM (Gibco) supplemented with 0.05% penicillin–streptomycin (Gibco) in a humidified incubator maintained at 37 °C and 5% $CO_2$. The human or mouse constructs encoding *KCNA2* were subcloned into the pXOOM vector using Not1 and HindIII restriction sites. pXOOM plasmids encoding human or mouse wild-type *KCNA2* and its variants were transfected using the Effectene transfection reagent (Qiagen) according to the manufacturer's protocol. Transfected cells were plated on 12 mm round coverglass 6–8 h before recording. Whole-cell patch clamp was performed using a custom electrophysiology rig. An Olympus CKX41 inverted microscope equipped with bandpass filters and a mercury arc lamp source was used to visualize EGFP-expressing cells, which were indicative of KCNA2 expression. Patch pipettes (3–5 MΩ resistance) freshly pulled from glass capillaries using flaming/brown P-97 pipette puller (Sutter) were maneuvered into place using a micromanipulator (MP-225; Sutter). Break-in was achieved post Giga-ohm seal establishment, and the cells were held at −60 mV. Step depolarizing potentials of 500 ms were applied every 10 mV from −60 mV to elicit potassium currents. Stimulus and recording were commanded using Patch-master (HEKA) on a UPC-10 amplifier (HEKA) and saved for offline analysis using Fitmaster (HEKA) and Origin (OriginLab). The experiments for each construct were performed over at least two different transfections over multiple days and early cell passages.

## Plasmid biosensor construction and transfection protocols

Site-directed mutagenesis was employed to introduce mutations into *KCNA2* using the QuikChange Site-Directed Mutagenesis Kit (Stratagene). This was performed on the construct containing human *KCNA2* and a rat *KCNA2* construct with an N-terminal EGFP tag in the pEGFP-C1 vector, featuring an extracellular HA site between the S1 and S2 transmembrane helices. H9c2 and COS-7 cells were cultured in a humidified incubator at 37 °C with 5% $CO_2$. These cells were grown in Dulbecco's modified Eagle's medium (DMEM, Gibco,12800017) supplemented with 10% FBS (Gibco Cat no. A5256701) and 1% penicillin/streptomycin (Gibco,15140122). Transfection with KCNA2 (WT) and variant constructs was performed for 48 h using jetPRIME (Polyplus Transfection, USA).

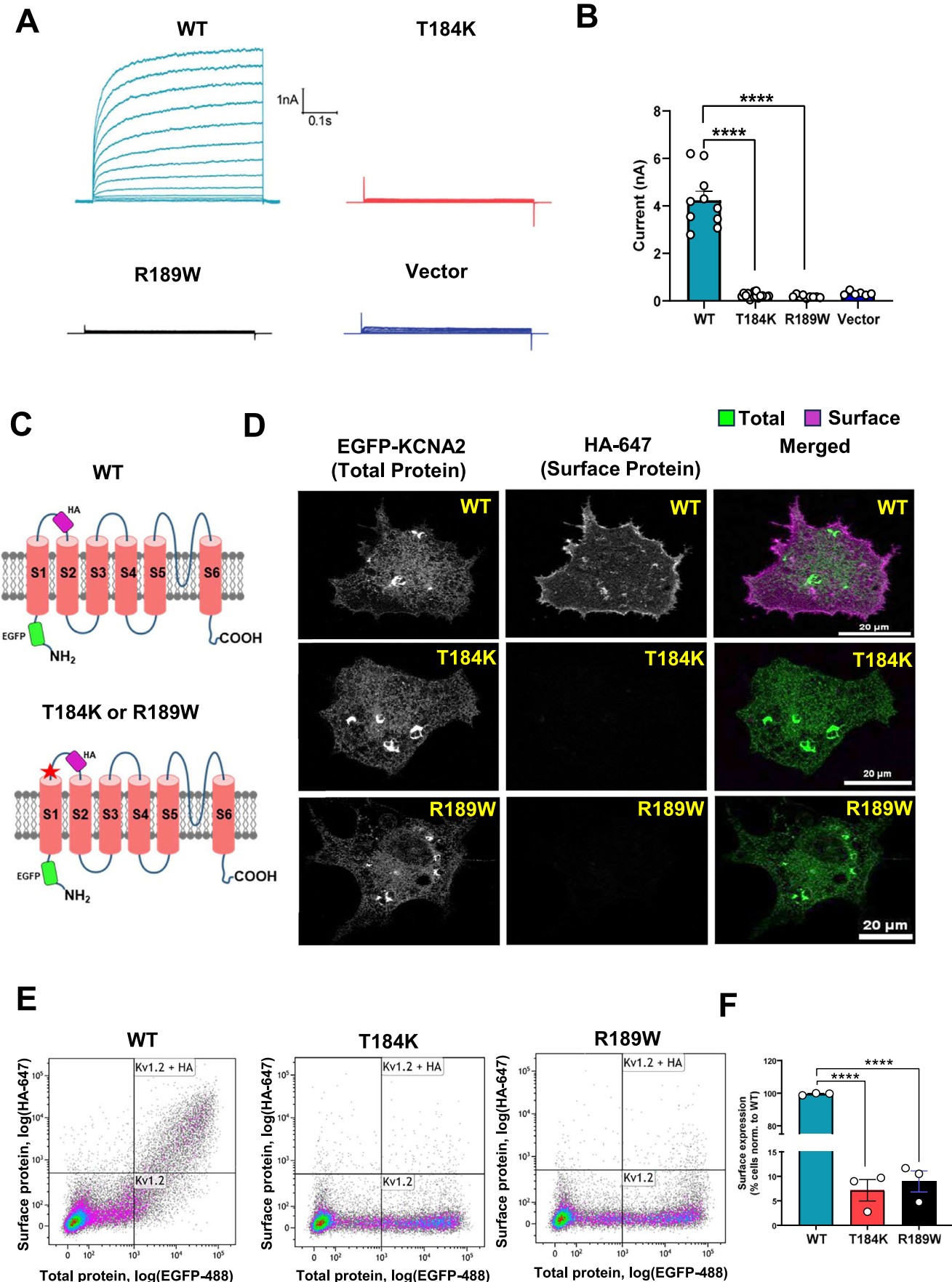

◄ **Figure 2. KCNA2 variants display defective voltage-dependent currents and cell surface trafficking.**

(A) Whole-cell patch-clamp analysis of KCNA2 wild-type (light blue), T184K (red), R189W (black), and empty pXoom vector (dark blue). Representative currents (nA) were elicited by step depolarization of HEK293T cells expressing the heterologous channels. (B) Total currents (mean ± SEM) recorded from each channel at a potential of +60 mV are presented. (WT $n = 10$, T184K $n = 16$, R189W $n = 9$, Vector $n = 6$ cells) Significance was evaluated by one-way ANOVA with post hoc Dunnett's multiple comparisons test ****$P < 0.0001$. WT vs T184K, $P = 2.2E-17$; WT vs R189W, $P = 1.7E-16$. (C) KCNA2 constructs used for trafficking studies in (D–F). (D) Confocal micrographs of non-permeabilized COS7 expressing the indicated KCNA2 constructs. Scale bar = 20 μm. (E) Flow cytometric analysis of live COS7 cells expressing the indicated KCNA2 constructs. Plots representing cell density of log(EGFP) (total KCNA2) against log(HA) (KCNA2 on the cell surface). The vertical line separates the low- and high-EGFP cells, and the horizontal line separates the α-HA-negative and -positive cells (WT $n = 7670$, T184K $n = 7349$, R189W $n = 7441$, KCNA2-positive cells). (F) The percentage of cells showing cell surface expression of KCNA2 (WT or variants) normalized to that of WT. The experiment was performed in biological triplicate ($n = 3$). The values indicate the mean ± SEM. Significance was evaluated using one-way ANOVA with post hoc Dunnett's multiple comparisons test ****$P < 0.0001$. WT vs T184K, $P = 2.5E-10$; WT vs R189W, $P = 3.2E-10$. Source data are available online for this figure.

For RAC1 inhibition studies in H9c2 cells, we utilized a well-known RAC1 inhibitor EHT1864 (Shutes et al, 2007) (Cayman Chemical, 17258). At 40 h post-transfection, the cells were cultured in DMEM with 2% FBS for 6 h before treatment with 10 μM EHT1864 for 2 h (Xu et al, 2020). In addition, Simvastatin, which also significantly inhibits RAC1 through pleiotropic effects, was used as a RAC1 inhibitor (RACi) (Chen et al, 2008; Takemoto et al, 2001) (Cayman Chemical, 10010344). The cells were exposed to 12 μM simvastatin for 24 h after 24 h of transfection (Gbelcová et al, 2013; Sundararaj et al, 2008). The cell lines were tested negative for mycoplasma and were authenticated at source with STR profiling.

## Cell surface trafficking of KCNA2

For immunocytochemistry (ICC), 15,000 COS-7 cells were placed in each 35 mm glass-bottom dish (Ibidi, 81218-200) and transfected with 1 μg EGFP-KCNA2-HA constructs, including both wild-type and variants, for 48 h using Jetprime polyplus (Sartorius). Forty-eight hours post-transfection, the cells were washed twice with PBS, fixed with 4% PFA for 15 min, and then washed thrice with PBS. Blocking was performed for 1 h using 5% neonatal goat serum(NGS) in PBS. Subsequently, the cells were incubated overnight at 4 °C with 5 μg/ml of HA conjugated with Alexa Fluor 647 (BioLegend, 682404) in the blocking buffer. Following nuclear staining with Hoechst (Thermo Fisher Scientific, 62249) at a 1:10,000 dilution for 5 min, the cells were washed thrice with PBS. Imaging was performed using an Olympus Fv3000 confocal microscope, and the images were analyzed using ImageJ software.

## Flow cytometry analysis

In the flow cytometry experiment, 30,000 COS-7 cells were seeded in each well of a 24-well plate and transfected with 1 μg of EGFP-KCNA2-HA constructs, including both wild-type and variants, for 48 h. After transfection, the cells were detached using trypsin (Thermo Fisher Scientific, 25200056) and were washed with PBS. The samples were then centrifuged at $400 \times g$ for 5 min at 4 °C. The cells were initially stained with Zombie Violet viability dye (1:100) (BioLegend, 423113) for 15 min. Subsequently, they were stained with anti-HA conjugated with Alexa Fluor 647 at a concentration of 8 μg/ml (BioLegend, 682404) in PBS containing 5% FBS for 45 min at 4 °C. Following three PBS washes, the cells were analyzed using BD LSRFortessa™ X-20. Kaluza analysis software (Beckman Coulter Life Sciences) was used for further data analysis. Zombie violet staining marked the dead cells that were gated out, and only live cells were used for the analysis.

## Whole-cell protein extraction and immunoblotting

The cells were lysed using RIPA Buffer (Sigma-Aldrich, R0278), with 1 mM PMSF (Thermo Fisher Scientific, 36978) and protease and phosphatase inhibitor cocktail (Thermo Fisher Scientific, 78440). A total of 15 μg of protein was loaded for the SDS-polyacrylamide gel electrophoresis. Protein gels were transferred onto polyvinylidene fluoride (PVDF) membranes (BioRad, 1620177) as per the standard protocol. The membranes were then blocked with 3% BSA and incubated with primary antibodies overnight at 4 °C. The primary antibodies used at a 1:1000 dilution included extracellular signal-regulated kinase 1/2 (ERK1/2, Cell Signalling Technologies, 9102S), phospho-ERK1/2(Thr202/Tyr204, Cell Signalling Technologies, 9101S), glyceraldehyde-3-phosphate dehydrogenase (GAPDH) (Thermo Fisher Scientific, MA5-15738), RAC1 (Cell Signalling Technologies, 8631), and KCNA2 (Kv1.2, Abcam, ab192758). The membranes were then treated with suitable HRP-conjugated secondary antibodies (anti-mouse: 31430, Thermo Fisher Scientific, and anti-rabbit: 31460, Thermo Fisher Scientific), and the signal intensities were detected using enhanced chemiluminescence (Cyanagen). GAPDH was used as the internal loading control. ImageJ software was used for the analysis.

## Immunocytochemistry (ICC)

Cardiomyocytes, including H9c2 and iPSC-derived cardiomyocytes (P2 and P3) as well as iPSC colonies, were seeded in 35-mm glass-bottom culture dishes (Ibidi, 81218-200). The cells were rinsed twice with PBS before being fixed in 4% PFA for 15 min. Following another PBS wash, the cells were permeabilized with 0.1% TritonX-100 (Sigma-Aldrich, X100) in PBS for 15 min. The cells were then blocked using ICC blocking buffer consisting of 2% BSA (HiMedia, MB083) and 2% NGS in PBS, for 1 h. Subsequently, the cells were incubated overnight at 4 °C with their specific primary antibodies: α-sarcomeric actinin (Dilution 1:250 - 701914, Thermo Fisher Scientific), Nanog (Dilution 1:200—Santa Cruz, sc-293121), TRA-1-81 (Santa Cruz, sc-21706), and SSEA4 (Dilution 1:200—Santa Cruz, sc-59368) in ICC blocking buffer. After washing with PBS, the cells were incubated with their respective secondary antibodies (A-11008, A-11003, Thermo Fisher Scientific) and Phalloidin Alexa Fluor 546 (dilution 1:500) (Thermo Fisher Scientific, A22283) for 1 h. Hoechst (Thermo Fisher Scientific, 62249) was used at a 1:10,000 dilution for nuclear staining. The cells were washed thrice with PBS, each wash lasting 5 min, with gentle agitation. Imaging was performed using an Olympus Fv3000 confocal microscope or an Olympus IX73 microscope, and the images were analyzed using ImageJ software.

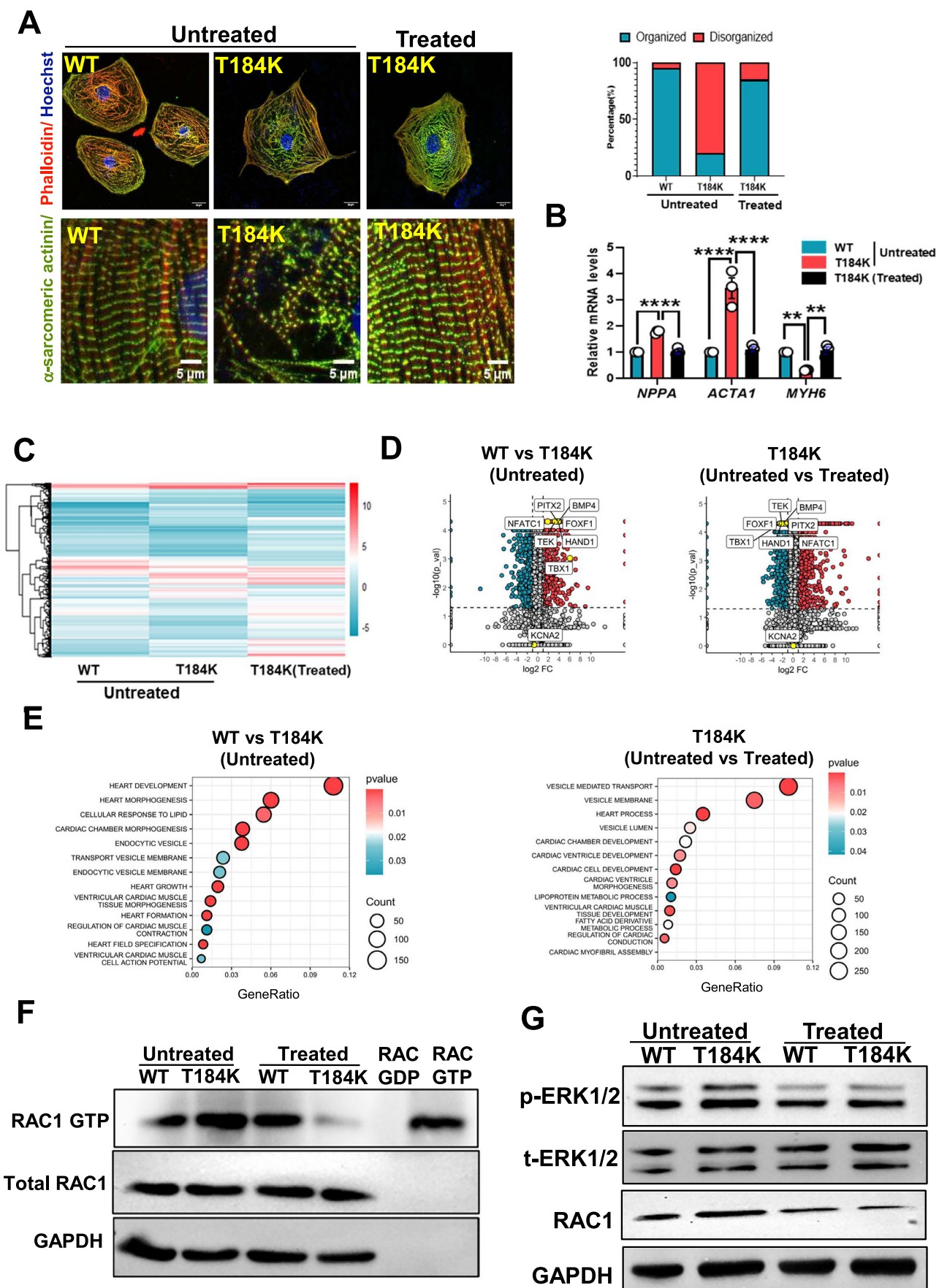

◄ **Figure 3. KCNA2 p.T184K iPSC-CMs display DCM phenotypes that are rescued upon simvastatin treatment.**

(A) Confocal micrographs of 30-day-old iPSC-CMs Control (WT) and KCNA2 p.T184K (T184K) with untreated and treated (1 μM simvastatin) conditions for 48 h stained with phalloidin (red), α-sarcomeric actinin (green), and Hoechst (blue), and micrographs showing myofibril organization. Scale bars = 20 μm and 5 μm from top to bottom. Percentage of cells showing disorganized myofibrils in WT (untreated) and p.T184K iPSC-CMs (untreated and treated) (n = 20 cells per group). (B) Quantitative real-time PCR analysis of heart failure markers *NPPA, MYH6, and ACTA1* in WT and T184K iPSC-CMs in the indicated conditions. The mRNA levels were normalized to *RNU6-1* and are presented as relative expression levels (n = 3). The values indicate means ± SEM. Significance was evaluated using two-way ANOVA with post hoc Tukey's multiple comparisons test. **$P < 0.01$, ****$P < 0.0001$. The values are from biological replicates represented as means ± SEM. *NPPA*(WT vs T184K untreated, $P = 0.0041$, *T184K* untreated vs. T184K treated, $P = 0.0050$), *ACTA1* (WT vs T184K untreated, $P = 1.8E-9$, T184K untreated vs. T184K treated, $P = 3.9E-9$) *MYH6* (WT vs T184K untreated, $P = 0.0099$, T184K untreated vs. T184K treated, $P = 0.0052$). (C) Heatmap of transcriptomes comparing the log₂(FPKM) between untreated (WT and T184K) and treated (T184K) iPSC-CMs (total of 1883 dysregulated genes). (D) Volcano plots depicting heart failure genes being upregulated in the T184K iPSC-CMs, rescued upon treatment with simvastatin (a total of 3675 dysregulated genes in WT vs T184K and 5502 dysregulated genes in T184K vs T184K treated cardiomyocytes). Pink: upregulated genes, blue: downregulated genes, gray: unchanged, yellow: highlighted genes. The data represented is from three independent experiments assayed in duplicates. Statistical tests were carried out using Cuffdiff likelihood ratio test (LRT) with Benjamini–Hochberg false discovery rate (FDR) correction. (E) Dot plots of the gene ontology significant processes (KCNA2 related pathways) in untreated WT vs T184K(left) and untreated vs treated (T184K) (right) iPSC-CMs. (F) Immunoblots of RAC1 activity in the indicated conditions, along with the total lysate. (G) Immunoblots of the whole-cell lysates with the indicated proteins from the 30-day-old untreated and treated (48 h) WT and T184K iPSC-CMs. Source data are available online for this figure.

## Real-time quantitative PCR analysis

RNA was extracted using TRIzol (Thermo Fisher Scientific, 423113), and cDNA was synthesized using the Verso cDNA Synthesis Kit (Thermo Fisher Scientific, AB1453A). cDNA was quantified using real-time PCR with PowerUp™ SYBR™ Green Master Mix (Thermo Fisher Scientific, A25742) on an ABI QuantStudio 5 real-time PCR system (Thermo Fisher Scientific). The relative abundance of mRNA was determined as a fold change compared to 18S-rRNA or *RNU6-1*, which served as the housekeeping gene, and was normalized to the wild-type or control group.

## RAC1 activity assay

RAC1 activity was assessed using the Active RAC1 Detection Kit (Cell Signalling Technologies, 8815) according to the manufacturer's guidelines. In brief, whole-cell lysates (500 μg) were incubated with 20 μg of GST-Human PAK1-PBD in a spin cup containing glutathione resin and incubated at 4 °C for 1 h. Following incubation, the spin cup was centrifuged at 6000×g for 30 s. The resin was washed twice with the cell lysis buffer included in the kit. A reducing buffer was prepared by adding dithiothreitol (DTT) at 200 mM in 2× SDS Sample Buffer. Each column was incubated with 50 μL of the reducing sample buffer for 2 min at room temperature, followed by centrifugation at 6000×g for 2 min. The eluted samples were then heated at 95 °C for 5 min, and the RAC1 GTP levels in these samples were measured using immunoblotting. For the controls, the whole-cell lysate was incubated with either 0.1 mM GTPγS for a positive control, i.e., RAC1 Guanosine triphosphate (RAC1GTP) or 0.1 mM GDP for a negative control, i.e., RAC1 Guanosine diphosphate (RAC1GDP).

## iPSC generation and characterization

Peripheral blood mononuclear cells (PBMCs) from patient (P3) and familial WT control (P2) were reprogrammed using the CytoTune-iPS 2.0 Sendai virus-based reprogramming kit (Thermo Fisher Scientific, A16517) following the manufacturer's guidelines and characterized as previously described by us (Chimata et al, 2022; Jain et al, 2025). In brief, PBMCs were cultured in PBMC medium (StemPro™ -34 (Thermo Fisher Scientific, 10639011) with Glutamax/2 mM L-Glutamine, supplemented with cytokines: SCF

at 100 ng/mL (Thermo Fisher Scientific, PHC2115), FLT-3 at 100 ng/mL (Thermo Fisher Scientific, PHC9414), IL-3 at 20 ng/mL (Thermo Fisher Scientific, PHC0034), and IL-6 at 20 ng/mL (Thermo Fisher Scientific, PHC0065) using an ultra-low attachment 24-well plate for 4 days. The PBMCs were transduced with Sendai viruses at the following MOIs: KOS (virus containing Klf4, Sox2, and Oct3/4) with MOI = 5, hc-Myc (virus containing c-Myc) MOI = 5, hKlf4 (virus containing Klf4) MOI = 3. After 24 h, the medium containing the viral remnants was replaced with fresh PBMC medium. After 48 h, the cells were transferred to vitronectin-coated (Thermo Fisher Scientific, A14700) dishes with StemPro™-34 media, which was replaced every other day. From day 7, the medium was transitioned to Essential 8 (Thermo Fisher Scientific, 10639011), and iPSC colonies began to appear from day 10 onwards. Subsequently, the iPSCs were cultured in StemFlex medium (Thermo Fisher Scientific, A3349401).

Embryoid bodies were generated as described previously (Jain et al, 2025). Briefly, iPSC cell aggregates, formed after ReLeSR™ (Stem Cell Technologies, 100-0483) treatment, were cultured in a low-attachment dish containing StemFlex medium for 72 h to form embryoid bodies. Embryoid bodies were assessed for tri-lineage markers using qRT-PCR to evaluate their pluripotency.

The iPSC colonies were stained to detect stemness markers, including NANOG, TRA-1-81, and SSEA4 as described previously (Chimata et al, 2022; Jain et al, 2025). In addition, RNA was extracted from the iPSCs, and reverse transcription PCR was performed to confirm the absence of Sendai Virus, following the CytoTune-iPS 2.0 manual, using a Sendai positive control and RNA from H9 human embryonic stem cells as a negative control.

## Differentiation of iPSCs into cardiomyocytes

To induce the differentiation of iPSCs into cardiomyocytes, on day 1, iPSCs were cultured in cardiac differentiation media, which consisted of RPMI (Thermo Fisher Scientific, 72400047) supplemented with 2% B27 minus insulin (Thermo Fisher Scientific, A1895601) containing 6 μM CHIR99021 (Sigma-Aldrich, SML1046). The cells were cultured in this medium for 24 h, following which they were replaced with cardiac differentiation media. On days 3 and 4, the cells were exposed to 5 μM IWP2 (Sigma-Aldrich, I0536) in the cardiac differentiation media. The culture was replaced on day 5 with cardiac differentiation media

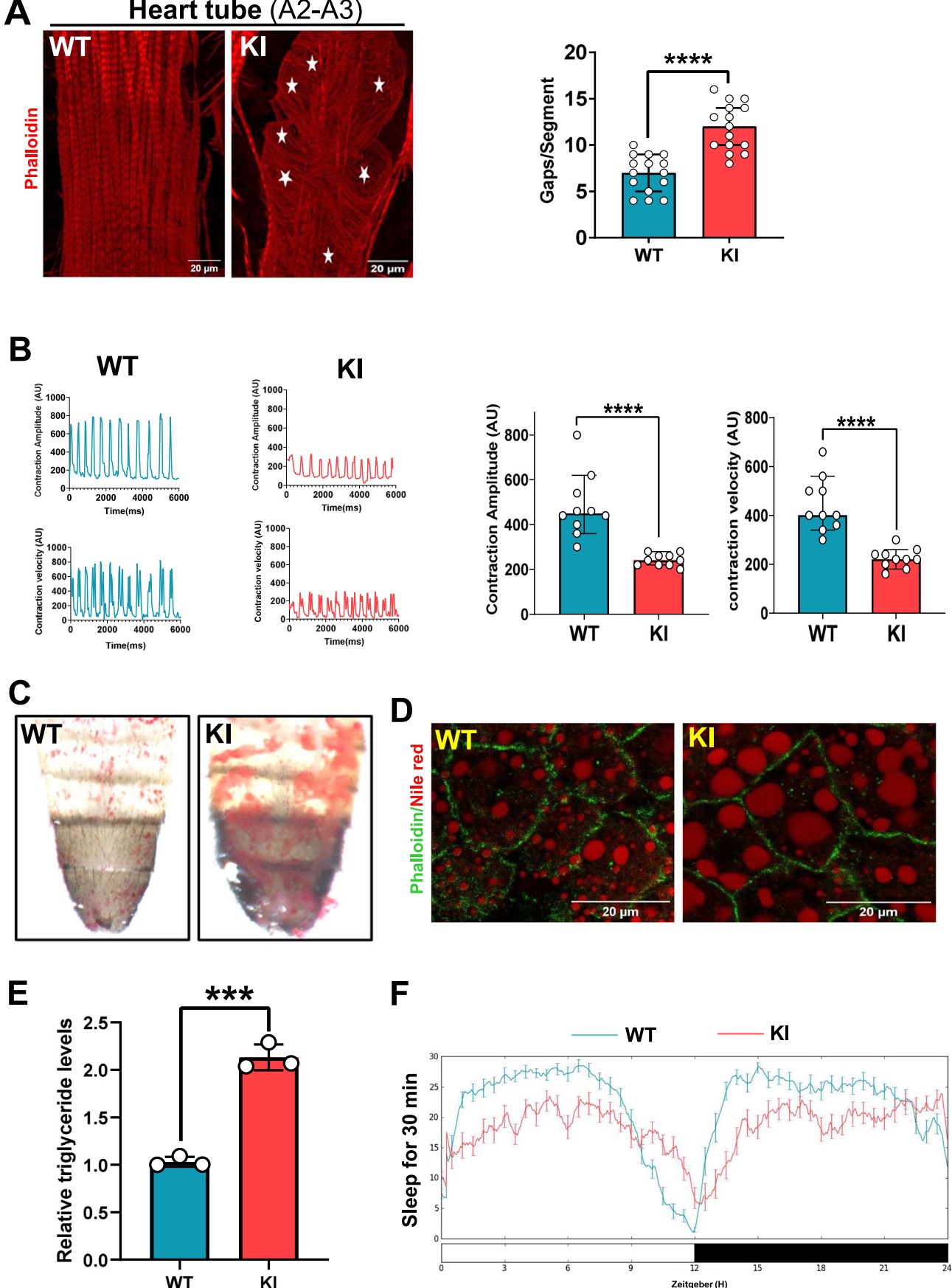

**Figure 4. KCNA2 variant displayed DOSA-like phenotypes in the *Drosophila* model.**

(A) *Drosophila* heart tube (Abdominal segment (A2-A3)) stained with phalloidin (red), showing myofibrillar disorganization (left) and its quantification (right). The stars (white) indicate the gaps in the tube ($n = 15$ in each group). The values represented are means ± SEM. Significance was evaluated using an unpaired two-tailed Student's *t* test. ****$P < 0.0001$. WT vs KI, $P = 2.0E{-}6$. Scale bar = 20 μm. (B) Heart tube contraction amplitude (top) and velocity (bottom) traces from live semi-intact *Drosophila* heart tubes and their quantifications obtained through analysis of videos using MUSCLEMOTION ($n = 10$ in each group). The values represented are means ± SEM. Significance was evaluated using unpaired two-tailed Student's *t* test. ****$P < 0.0001$. Contraction amplitude (WT vs KI, $P = 5.7E{-}5$), contraction velocity (WT vs KI, $P = 1.5E{-}5$). (C) Abdominal fat stained for Oil Red O (red). (D) Confocal micrographs of the abdominal fat bodies from WT and KI stained for Nile red, with the fat body (red) and Phalloidin (green). Scale bar = 20 μm. (E) Triglyceride levels of the KI *Drosophila* normalized to the WT using the triglyceride assay ($n = 3$ biological repeats each for WT and KI). The values represented are means ± SEM. Significance was evaluated using unpaired two-tailed Student's *t* test (WT vs KI). ***$P < 0.001$. WT vs KI, $P = 0.0002$. (F) Representative sleep profile of the WT and KI *Drosophila* in a 12-h dark and light cycle. (WT $n = 30$, KI $n = 22$) flies per group. The values are from biological replicates represented as means ± SEM. Source data are available online for this figure.

and maintained until day 7, with daily media changes. On day 8, the culture was supplemented with cardiac maintenance media (RPMI containing 2% B27 supplement, Thermo Fisher Scientific, 17504044), and the media was replaced every 2 days until day 28, at which point the cells were treated with 1 μM Simvastatin for a duration of 48 h. An iPSC-cardiomyocyte is considered disorganized or disordered when the sarcomeric structure is visually misaligned.

## Generation of cardiac organoids

Cardiac organoids were generated using the protocol followed previously (Lewis-Israeli et al, 2021). Both the control and the patient-derived iPSCs were dissociated into individual cells using StemPro Accutase (Thermo Fisher Scientific, A1110501) and seeded at a density of 10,000 cells per well in 100 μl of StemFlex media in an ultra-low attachment round-bottom 96-well plate (Corning, Costar, 7007), along with RevitaCell Supplement (Thermo Fisher Scientific, A2644501). The plate was centrifuged at $100{\times}g$ for 3 min and was incubated at 37 °C for 24 h to generate embryonic bodies. The next day (day −1), 50 μl of the spent media was replaced with 200 μl of fresh StemFlex media, resulting in a total volume of 250 μl. On day 0, differentiation was initiated using 4 μM CHIR99021 (Sigma-Aldrich, SML1046), 1.25 ng/ml BMP4 (Sigma-Aldrich, GF302), and 1 ng/ml Activin A (Sigma-Aldrich, SRP3003) in cardiac differentiation media (RPMI with 2% B27 supplement minus insulin) and incubated for 24 h, followed by maintenance in the cardiac differentiation media for another 24 h. On day 2, cardiac mesodermal specification was induced with 2 μM Wnt-C59 (Sigma-Aldrich, AMBH2D6FA48B) in cardiac differentiation media and incubated for 48 h, after which the media was replaced with fresh cardiac differentiation media for another 48 h without any media change. From day 6 onwards, the cardiac differentiation media were switched to cardiac maintenance media (RPMI with 2% B27 supplement) for 24 h. On day 7, the epicardial lineage was induced with 2 μM CHIR9902 for 1 h. The cardiac maintenance media were subsequently refreshed every 48 h until day 21.

## Global RNA-seq analysis

RNA was extracted from 30-day-old WT and p.T184K iPSC cardiomyocytes (untreated and treated). RNA integrity was evaluated using TapeStation (Agilent). The RNA library was constructed using the NEBNext® Ultra™ II Directional RNA Library Prep with Sample Purification Beads (Catalog no-

E7765L). Sequencing was performed on the NovaSeq 6000 platform, with a read length of 2×100 bp, yielding ~30–35 million reads per sample. Sequencing data were trimmed for adapter sequences and low-quality reads using Trim Galore! and were aligned to the GRCh38 genome using TopHat2. Paired-end reads from both ends of the same RNA fragment were analyzed, and paired reads ≥ 35 bp were retained. Transcript assembly was performed using Cufflinks, transcripts were combined using cuffmerge, and differentially expressed genes (DEGs) were analyzed using Cuffdiff. Statistical analysis and visualization were carried out using the CummeRbund package in R. Finally, gene ontology analysis was conducted using the ClusterProfiler package in R.

## *Drosophila*-related studies

*Drosophila* were reared on standard cornmeal medium at 25 °C. Canton S stock was used as the wild-type (WT) background. For this study, we employed *Drosophila* with a representative KCNA2 p.T184th mutation (Bloomington stock No.: 24149), referred to here as KI. In experiments studying triglyceride levels and fat bodies, *Drosophila* were reared on a high-fat diet containing 7% coconut oil (Sigma-Aldrich, C1758) at 21 °C. Fly eggs were laid on food that included either DMSO (Vehicle control) or 240 μM simvastatin (Cayman Chemical, 10010344), and the 5-day-old males that emerged from these vials were examined.

## Semi-intact *Drosophila* heart preparation and quantification of heartbeat rate

The 5-day-old male *Drosophila* were anesthetized by briefly exposing them to ice, positioned on a petri dish with their dorsal side facing down, and dissected following the previously outlined protocol guidelines (Vogler and Ocorr, 2009). The heart tubes were immersed in freshly aerated artificial hemolymph, as described, and videos were captured using a Nikon SMZ800N microscope. Heartbeat quantification was performed using the MUSCLEMOTION (Sala et al, 2018) software plugin in ImageJ.

## Immunostaining of *Drosophila* heart tubes

The heart tubes of *Drosophila* were fixed with 4% PFA for 15 min, followed by three washes with PBS. They were then permeabilized with 0.1% TritonX-100 in PBS for another 15 min. After washing three more times with PBS, the heart tubes were stained with phalloidin conjugated with Alexa Fluor 546 (Invitrogen) at a 1:200 dilution for 1 h at room temperature. Imaging was performed using

**The paper explained**

**Problem**

Dilated cardiomyopathy (DCM) is a heart condition characterized by impaired systolic and diastolic functions. Individuals with DCM frequently suffer from obesity and sleep apnea. Whether the genetic variations can cause combined phenotypes of DCM, obesity, and sleep apnea remains unclear.

**Results**

Here, we utilized next-generation sequencing and analysis of exomes from Indian patients and identified a pathogenic *KCNA2* variant in a patient exhibiting DCM, Obesity, and Sleep Apnea (termed DOSA). Functional analysis using electrophysiology and biosensors in cells expressing the KCNA2 variants revealed a loss of detectable whole-cell current due to trafficking defects. Also, cellular models, including 2D and 3D patient-specific iPSC cardiomyocytes and organoid models, demonstrated a role for RAC1-ERK1/2 in disease pathogenesis. A *Drosophila* model with a representative *KCNA2* mutation exhibited DOSA-like phenotypes via RAC1-ERK1/2 signaling activation, which was alleviated by RAC1 inhibitors.

**Impact**

Our findings provide the first evidence that *KCNA2* variants can lead to combined clinical phenotypes including DCM, obesity, and sleep apnea. These results further expand the genetic regulatory roles of potassium channels in human diseases.

an Olympus FV3000 confocal microscope, and the images were analyzed using ImageJ software.

## Triglyceride assay

In total, 5–8 male *Drosophila* (without head) per group were homogenized in 200 μl PBST (0.1% Tween 20 in PBS). The mixture was then heated at 70 °C for 5 min, followed by centrifugation at 4000 rpm for 5 min at 4 °C. From each sample, 50 μl of the supernatant was utilized to measure triglycerides using the Triglyceride Quantification Colorimetric/Fluorometric Kit from Sigma-Aldrich (TR0100). The results were normalized to the protein content of the extract, which was determined using the Pierce BCA Protein Estimation Kit (Thermo Fisher Scientific, 23225).

## *Drosophila* fat body staining

The 5-day-old male *Drosophila* were anesthetized by briefly exposing them to ice, positioned on a petri dish containing PBS with their dorsal side facing down. A vertical cut was made along the ventral side of the abdomen, and the internal organs were cleared out. The abdomens were subsequently fixed in 4% PFA for 20 min. These abdomens were subsequently used for either Nile Red or Oil Red O staining.

The fixed abdomens were stained with Nile Red (Thermo Fisher Scientific, N1142) at a concentration of 10 μg/ml along with Alexa 647-conjugated Phalloidin (Invitrogen) at a 1:200 dilution overnight at 4 °C. They were washed three times with PBS for 15 min each and then mounted on a slide for imaging using a confocal microscope (Olympus FV3000).

For Oil Red O staining, 0.1% Oil Red O (Sigma-Aldrich, O1391) solution was prepared in 60% isopropanol in water. The abdomens were stained with the above solution and incubated for 20 min. They were washed thrice with PBS for 15 min each and then mounted on a slide for imaging using a Nikon SMZ800N microscope.

## *Drosophila* activity assays

Five-day-old *Drosophila* were anesthetized using $CO_2$ pads and placed individually into glass tubes (5 mm), each containing food at one end. The tubes were then positioned in the slots of a *Drosophila* activity monitor (DAM) from Trikinetics. The activity of the flies was tracked at 1-min intervals, with sleep defined as a period of inactivity lasting 5 min or longer (Cirelli and Bushey, 2008). The incubator was programmed to follow a 12-h light and 12-h dark cycle. Data collection for the analysis commenced at Zeitgeber time 0 (ZT0). The gathered data were analyzed using the PySolo software (Gilestro and Cirelli, 2009).

## Statistical analysis

Statistical analyses, such as two-tailed unpaired Student's *t* tests, multiple *t* tests with post hoc Holm–Sidak method, one-way ANOVA with Tukey's multiple comparisons post hoc tests, two-way ANOVA with Tukey's multiple comparisons or Sidak's multiple comparisons post hoc tests. $P < 0.05$ was considered to be statistically significant. Two independent authors confirmed the sample analysis with a blinded approach. The exact *P* values are reported in the Appendix Table S1.

Lists of primers used in the study have been provided in the Appendix Table S2.

# Data availability

The datasets produced in this study are available in the following databases: RNA-seq data is available at NCBI BioProject Submissions: BioProject ID PRJNA1370669.

The source data of this paper are collected in the following database record: biostudies:S-SCDT-10_1038-S44321-026-00391-y.

# Peer review information

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

## Acknowledgements

PSD is supported by the Department of Biotechnology (DBT) (BT/PR45262/MED/12/955/2022, Anusandhan National Research Foundation (ANRF) (CRG/2023/004193), Indian Council for Medical Research (ICMR) IRIS Id–2023-19831, and Indo-French Centre for the Promotion of Advanced Research (IFCPAR/CEFIPRA) CSRP project no. 7003-1. PSD is a recipient of the American Heart Association (AHA) International Professor award. The Estonian team acknowledges the Estonian Centre of Genomics/Roadmap II, funded by the Estonian Research Council (project number TT17), for their work. We also thank the Estonian Biobank research team, including Andres Metspalu, Lili Milani, Tõnu Esko, Reedik Mägi, Mait Metspalu, Mari Nelis, and Georgi Hudjashov. We thank Dr. Kashyap Krishnasamy and Vinay J Rao for the initial technical help in this study. We thank Dr. Antonios Pantazis for providing KCNA2 plasmids. We thank all participants for making this study possible. We acknowledge various Institutional core facilities for this work.

## Author contributions

**Prasanth Chimata**: Data curation; Formal analysis; Validation; Investigation; Methodology; Writing—original draft; Writing—review and editing. **Sahil Lall**: Methodology; cellular patch clamp studies. **Tarmo Annilo**: Resources. **M K Mathew**: Resources; Methodology; cellular patch clamp studies. **Andres Metspalu**: Resources. **Jayaprakash Shenthar**: Resources. **Perundurai S Dhandapany**: Conceptualization; Resources; Data curation; Formal analysis; Supervision; Funding acquisition; Validation; Investigation; Visualization; Methodology; Writing—original draft; Project administration; Writing—review and editing.

Source data underlying figure panels in this paper may have individual authorship assigned. Where available, figure panel/source data authorship is listed in the following database record: biostudies:S-SCDT-10_1038-S44321-026-00391-y.

## Disclosure and competing interests statement

The authors declare no competing interests.

