## [Peer Review File · EMBO Molecular Medicine]

KCNA2 variants cause dilated cardiomyopathy, obesity and sleep apnea through RAC-ERK pathway

Prasanth Chimata, Sahil Lall, Tarmo Annilo, M K Mathew, Andres Metspalu, Jayaprakash Shenthathar, and Perundurair S

Dhandapany

Corresponding author: Perundurair S Dhandapany (dhan@instem.res.in)

Review Timeline:

Submission Date:	3rd Sep 25
Editorial Decision:	23rd Sep 25
Revision Received:	12th Dec 25
Editorial Decision:	14th Jan 26
Revision Received:	6th Feb 26
Accepted:	10th Feb 26

Editor: Zeljko Durdevic

Transaction Report:

23rd Sep 2025

Dear Dr. Dhandapany,

Thank you for the submission of your manuscript to EMBO Molecular Medicine. We have now received feedback from the two reviewers who agreed to evaluate your manuscript. As you will see from the reports, both referees recognize potential interest of the study but also raise serious concerns that should be addressed in a major revision. If you would like to discuss further the points raised by the referees, I am available to do so via email or video. Let me know if you are interested in this option.

We would welcome the submission of a revised version within three months for further consideration. Please let us know if you require longer to complete the revision.

I look forward to receiving your revised manuscript.

Yours sincerely,

Zeljko Durdevic

Zeljko Durdevic
Senior Editor
EMBO Molecular Medicine

We require:

- 1) A .docx formatted version of the manuscript text (including legends for main figures, EV figures and tables). Please make sure that the changes are highlighted to be clearly visible.
- 2) Individual production quality figure files as .eps, .tif, .jpg (one file per figure). For guidance, download the 'Figure Guide PDF': (<https://www.embopress.org/page/journal/17574684/authorguide#figureformat>).
- 3) A .docx formatted letter INCLUDING the reviewers' reports and your detailed point-by-point responses to their comments. As part of the EMBO Press transparent editorial process, the point-by-point response is part of the Review Process File (RPF), which will be published alongside your paper.
- 4) A complete author checklist, which you can download from our author guidelines (<https://www.embopress.org/page/journal/17574684/authorguide#submissionofrevisions>). Please insert information in the checklist that is also reflected in the manuscript. The completed author checklist will also be part of the RPF.
- 5) Please note that all corresponding authors are required to supply an ORCID ID for their name upon submission of a revised manuscript.
- 6) It is mandatory to include a 'Data Availability' section after the Materials and Methods. Before submitting your revision, primary datasets produced in this study need to be deposited in an appropriate public database, and the accession numbers and

database listed under 'Data Availability'. Please remember to provide a reviewer password if the datasets are not yet public (see <https://www.embopress.org/page/journal/17574684/authorguide#dataavailability>).

12) Author contributions: You will be asked to provide CRediT (Contributor Role Taxonomy) terms in the submission system. These replace a narrative author contribution section in the manuscript.

13) A Conflict of Interest statement should be provided in the main text.

14) Every published paper now includes a 'Synopsis' to further enhance discoverability. Synopses are displayed on the journal webpage and are freely accessible to all readers. They include a short stand first (maximum of 300 characters, including space) as well as 2-5 one-sentences bullet points that summarizes the paper. Please write the bullet points to summarize the key NEW findings. They should be designed to be complementary to the abstract - i.e. not repeat the same text. We encourage inclusion of key acronyms and quantitative information (maximum of 30 words / bullet point). Please use the passive voice. Please attach these in a separate file or send them by email, we will incorporate them accordingly.

15) Include a Reagents and Tools Table as part of the Methods section, which can be downloaded from our author guidelines (<https://www.embopress.org/page/journal/17574684/authorguide#structuredmethods>)

***** Reviewer's comments *****

Referee #1 (Remarks for Author):

In this manuscript, Prasanth et al. investigated the role of KCNA2 mutations in the development of dilated cardiomyopathy, obesity, and sleep apnea (DOSA). The study is comprehensive and methodologically rigorous. The researchers identified two KCNA2 mutations-p.T184K in South Asian Indian families and p.R189W in Estonian Biobank-and characterized the clinical features of individuals carrying pathogenic variants of KCNA2. The involvement of DOSA-related KCNA2 mutations in iPSC-CMs and organoid models, as well as in transgenic Drosophila model, was clearly demonstrated, with particular emphasis on the activation of the RAC1-ERK1/2 signaling pathway. Importantly, this activation was shown to be reversible through the use of RAC1 inhibitors. The study employed a variety of experimental approaches, and the overall findings are well-supported. Nevertheless, there are several points that the authors should consider for clarification or further investigation.

Major comments:

1. Figure 1A: The result lacks sufficient detail regarding the sequencing methodology and the specific KCNA2 mutants. A schematic model should also be included for the Estonian patient cohort to ensure consistency.
2. Figure S1A: Do the p.T184K and p.R189W mutations alter the molecular weight of the KCNA2 protein? If so, distinct bands should be visible. If not, the current protein bands cannot conclusively demonstrate the effect of these mutations on the RAC1-ERK1/2 pathway. Additionally, statistical analysis of RAC1 and KCNA2 protein expression levels is required. The same applies to Appendix Figure S3b.
3. Figure 3A: The iPSC-CMs derived from the p.T184K patient appear smaller in size. However, since the patient is diagnosed with dilated cardiomyopathy, one would expect cardiomyocyte hypertrophy as a compensatory mechanism. This discrepancy requires further explanation.
4. Figure 4: In the p.T184K knock-in Drosophila model, the overexpression efficiency of the mutant allele in the fly heart should be validated.
5. The p.T184K mutation is identified in a male patient, what is the gender of the individual carrying the p.R189W mutation? Furthermore, does the drosophila model of DOSA associated with the p.T184K mutation exhibit sex-specific phenotypes?
6. Are there any individuals who carry both the p.T184K and p.R189W mutations simultaneously? If so, how do their clinical phenotypes compare to those with only a single mutation?

Minor comments:

1. Figure 3A: The methodology should be clarified. How are ordered and disordered arrangements distinguished?
2. Appendix Figure S3: The concentration should be corrected from "EHT1864 10mm" to "EHT1864 10 mM" in the figure legend?
3. Appendix Figure S3b and Figure 3F: The last two components are unclear. Please provide a detailed explanation of their meaning.
4. Why were different treatment conditions used for EHT1864 and simvastatin? Please justify the selection of concentrations and treatment durations.
5. The methodology for 3D organoid culture should be described in detail in the Materials and Methods.
6. Lines 121-129: The results should be described based on its structural arrangement, rather than by omitting or skipping portions of the results.
7. Standardize the use of punctuation and spacing throughout the manuscript, such as lines 841, 309, and 293.
8. In Figure 4, clarify the definitions of panels A2 and A3.

Referee #2 (Comments on Novelty/Model System for Author):

Novelty is medium, representing extensive functional characterization of two missense variants in the S1-S2 loop of KCNA2. Methods are extensive and are well documented. Impact is medium, focusing on two specific variants; however, impact on RAC1-ERB1/2 signaling pathways raises potential for development of targeted interventions. Use of drosophila as a model organism for study of cardiac disease/cardiomyopathy is supported by existing literature, but chosen ex vivo experiments are not well justified in the manuscript relative to in vivo imaging options, such as OCT.

Referee #2 (Remarks for Author):

This paper identifies two rare (MAF<0.01%) variants in the potassium voltage-gated channel gene, KCNA2, in Indian and Estonian patients with DOSA, a constellation of defects including dilated cardiomyopathy (DCM), obesity, sleep apnea. Variants were identified through exome sequencing and were validated by targeted Sanger sequencing. The Indian variant was not present in the proband's unaffected mother; and the mutation status of his affected father (d. SCD) was unknown). The variant was absent from six reference datasets, including gnomAD/ExAC/1000 genomes, as well as several other regional genetic databases. The Estonian variant was present at very low MAF in gnomAD, but absent from the regional reference populations. Impacted residues for both variants were conserved and occupied similar positions within the S1-S2 extracellular loop. Functional data is provided across several models: (1) HEK293/COS7 cells overexpressing variant KCNA2 failed to correctly localize KCNA2 to the plasma membrane, resulting in failure of whole cell current generation by patch-clamp analysis. (2) Experiments in H9C2 cardiomyocytes demonstrated increased cell sizes (hypertrophy) with elevated RNA-level expression of heart failure markers and elevated levels of activated (phosphorylated) ERK1/2, rescuable by pharmacologic treatment with RAC1 inhibitors (EHT1864 and Simvastatin). (3) Patient-derived iPSC-CMs and 3D cardiac organoids showed sarcomere disorganization alongside similar hyperactivations of RAC1-ERK1/2 signaling. (4) The T184K variant was modeled in a Drosophila knock-in (KI) model, previously published in Nature in 2005 as mns (minisleep), a line characterized by shortened sleep cycles and reduced lifespans. No cardiac data was presented in the 2005 publication and the mns phenotype is noted to be recessive. KCNA2 is, however, notably located on the X chromosome in flies, in contrast to autosomal chromosome 1 in humans. In the reviewed study, KI heart tubes showed disorganized myofibers with abnormal contractility, alongside elevated abdominal fat/lipids/triglycerides and reduced sleep. These phenotypes were rescuable with treatment with RAC1 inhibitors.

Overall this is a well-written and presented manuscript with logical presentation of functional data supporting pathogenicity of the identified variants. Support for variant causation is bolstered by pharmacological rescue across several disease models alongside requisite rarity, conservation, and in silico predictions of protein impact for rare-disease models. Comments to address are listed below:

Major Comments:

1. It is not clear what other genes were analyzed in these families. For DCM, close to 20 other genes have definitive to strong evidence for causation by ClinGen criteria (PMID: 33947203), with several others linked to obesity-related DCM (NOX4, HTRA1, KLHL29, (PMID: 36547458). A table should be added to the supplementary materials documenting P/LP/VUS variants in these other relevant genes. Given availability of DNA, presence or absence of any segregating variants in individual P2 (Pedigree 1) should also be documented.
2. A major advantage of selecting Drosophila as a model system for the T184K variant is that the line has already been comprehensively described in the context of abnormal sleep phenotypes. Nevertheless, cardiac experiments have not been adequately justified. Limitations of the ex vivo contractility experiments should be discussed alongside why they were chosen over in vivo imaging modalities, such as optical coherence tomography. Heart tube lumen and wall cell sizes were not visualized, precluding assessment of dilated versus hypertrophic phenotypes.
3. Importantly, I see no mention as to how results of previous KI fly studies support or otherwise inform new findings reported in this manuscript. These data should be discussed alongside any considerations of dominant versus recessive inheritance of the fly and human phenotypes.
4. The authors should address whether any other DCM-relevant signaling pathways (AKT, JNK, etc) were considered alongside ERK1/2.

Minor Comments:

1. I found the blue color of table 1c and e difficult to read, particularly in print format. I suggest removing the color entirely.
2. Inclusion of REVEL scores in Figure 1e would be informative.

Response to reviewers

We thank the editor and reviewers for the insightful review of the manuscript. The review comments are in red and the answers are in black. The changes in the manuscript are highlighted in yellow.

Referee #1 (Remarks for Author):

In this manuscript, Prasanth et al. investigated the role of KCNA2 mutations in the development of dilated cardiomyopathy, obesity, and sleep apnea (DOSA). The study is comprehensive and methodologically rigorous. The researchers identified two KCNA2 mutations-p.T184K in South Asian Indian families and p.R189W in Estonian Biobank-and characterized the clinical features of individuals carrying pathogenic variants of KCNA2. The involvement of DOSA-related KCNA2 mutations in iPSC-CMs and organoid models, as well as in transgenic *Drosophila* model, was clearly demonstrated, with particular emphasis on the activation of the RAC1-ERK1/2 signaling pathway. Importantly, this activation was shown to be reversible through the use of RAC1 inhibitors. The study employed a variety of experimental approaches, and the overall findings are well-supported. Nevertheless, there are several points that the authors should consider for clarification or further investigation.

Major comments:

1. Figure 1A: The result lacks sufficient detail regarding the sequencing methodology and the specific KCNA2 mutants. A schematic model should also be included for the Estonian patient cohort to ensure consistency.

Thanks for the reviewer's comment. The revised manuscript now includes updated information including the primer specifics for targeted sequencing (revised manuscript (word format) Methods line no 282-284). The schematic process and methodology related to sequencing and analysis for the Estonian patient biobank were previously detailed in an earlier publication which is now referred in the revised manuscript (Mitt *et al*, 2017) (revised manuscript(word format) Methods line no 285-286).

2. Figure S1A: Do the p.T184K and p.R189W mutations alter the molecular weight of the KCNA2 protein? If so, distinct bands should be visible. If not, the current protein bands cannot conclusively demonstrate the effect of these mutations on the RAC1-ERK1/2 pathway. Additionally, statistical analysis of RAC1 and KCNA2 protein expression levels is required. The same applies to Appendix Figure S3b.

We appreciate the reviewer's feedback. We carefully looked into it and re-ran the blots. The blots confirmed KCNA2 variants do not change the molecular weight as can be seen in the appendix figures S3D and S4C. Consequently, as demonstrated in Figure 2, the effect of the

mutated protein can be attributed to a defect in cellular trafficking. This may represent a mechanism by which the RAC1-ERK1/2 pathway is activated (reviewed in Palamidessi *et al*, 2008). Also, multiple RAC1 inhibitors rescued the KCNA2 induced phenotypes including DCM, obesity and sleep apnea in multiple models confirming its indispensable role as shown in our figures (Fig. 3, Appendix Figs. S4, S5, S6, S7 and S12). The statistical analysis of RAC1 and KCNA2 protein expression levels are given in the Appendix figures S3 F-G, S4 E-F and S7 F and I.

3. Figure 3A: The iPSC-CMs derived from the p.T184K patient appear smaller in size. However, since the patient is diagnosed with dilated cardiomyopathy, one would expect cardiomyocyte hypertrophy as a compensatory mechanism. This discrepancy requires further explanation.

Thank you for pointing out the discrepancy. In Figure 3A, the representative image of the wild-type iPSC-cardiomyocyte is shown to be multinucleated, in contrast to the mutant iPSC cardiomyocytes. To ensure consistency, we have provided a comparable wild-type image at a similar scale. As anticipated, the cell size of the mutant iPSC-cardiomyocyte is larger than that of the wild type.

4. Figure 4: In the p.T184K knock-in *Drosophila* model, the overexpression efficiency of the mutant allele in the fly heart should be validated.

We appreciate the feedback provided. The expression level of KCNA2 mRNA in the *Drosophila* mutant remained unchanged compared to the wild type, as corroborated by the study conducted by Cirelli *et al.* (Nature, 2005), which is cited in the manuscript. The Kv1.2 region in humans, mice, and *Drosophila* is highly conserved. However, specific antibodies for *Drosophila* Kv1.2 were not available. Our attempts to identify any changes in protein levels using available human/mouse antibodies, which might cross-react, showed no significant differences between the wild type and mutant *Drosophila*.

5. The p.T184K mutation is identified in a male patient, what is the gender of the individual carrying the p.R189W mutation? Furthermore, does the *drosophila* model of DOSA associated with the p.T184K mutation exhibit sex-specific phenotypes?

Thank you for the comment. According to the regulatory guidelines of the Estonian biobank, if a genetic variant is identified in fewer than five samples, only non-personally identifiable data, such as variant allele frequencies, can be disclosed, while individual-level information must remain confidential. Consequently, the gender and other parameters of the Estonian patient harboring the KCNA2 p.R189W mutation cannot be disclosed (revised manuscript Fig. 1). Also, we identified another variant p.T184I in an Indian patient who is a male (updated in the revised manuscript (word format) line no 73-78 and Fig.1C), further strengthening our finding and

suggesting males are predominantly affected in this rare syndrome. The fly mutant demonstrates sex-specific phenotypes, with females exhibiting relatively milder cardiac and obesity phenotypes. However, we exclusively utilized male flies for our study.

6. Are there any individuals who carry both the p.T184K and p.R189W mutations simultaneously? If so, how do their clinical phenotypes compare to those with only a single mutation?

Thank you for the comments. We did not identify compound heterozygous mutations in the same patient; however, we discovered a rare homozygous mutation, p.T184I, in an independent Indian cohort of childhood dilated cardiomyopathy. The patient with the p.T184I mutation exhibited phenotypes similar to those of the p.T184K patient, but at an earlier age (line no 73-78 and Figure 1B&C in the revised manuscript (word format)). These data suggest that the homozygous variant results in earlier symptoms compared to carriers of the heterozygous variant.

Minor comments:

1. Figure 3A: The methodology should be clarified. How are ordered and disordered arrangements distinguished?

Thank you for your comment. An iPSC-cardiomyocyte is considered disorganized or disordered when its sarcomeric structure is visually misaligned. This detail has been revised in the methodology section (revised manuscript (word format) line no 446-447).

2. Appendix Figure S3: The concentration should be corrected from "EHT1864 10mm" to "EHT1864 10 mM" in the figure legend?

Thank you, we have corrected the same (Revised manuscript Appendix Figure S4).

3. Appendix Figure S3b and Figure 3F: The last two components are unclear. Please provide a detailed explanation of their meaning.

Thank you for the comment. In Appendix Figure S3b (Appendix Figure S4B in the revised manuscript) and Figure 3F, RAC1GDP (RAC1 Guanosine diphosphate) denotes the inactive form of RAC1, serving as a negative control. Conversely, RAC1GTP (RAC1 Guanosine triphosphate) signifies the active form of RAC1 and functions as a positive control. These data are updated in the revised methodology (revised manuscript(word format) line no 401-403).

4. Why were different treatment conditions used for EHT1864 and simvastatin? Please justify the selection of concentrations and treatment durations.

Thank you for your comment. The treatment parameters, including concentrations and durations, for EHT1864 and simvastatin varied according to the established protocols designed

to effectively inhibit RAC1 activity, whether directly or indirectly(Xu *et al*, 2020; Gbelcová *et al*, 2013; Sundararaj *et al*, 2008) (updated in the revised manuscript(word format) Methods line no 318-319 and line no 321-323)

5. The methodology for 3D organoid culture should be described in detail in the Materials and Methods.

Thank you. The 3D organoid culture methodology is detailed in the revised manuscript word format (revised manuscript(word format) Methods line 448-468).

6. Lines 121-129: The results should be described based on its structural arrangement, rather than by omitting or skipping portions of the results.

Thank you for your feedback. The manuscript has been updated accordingly (revised manuscript (word format) line no 125), and the relevant details are presented in Figure 3A. Unfortunately, we were unable to provide a more comprehensive explanation due to space limitations.

7. Standardize the use of punctuation and spacing throughout the manuscript, such as lines 841, 309, and 293.

Thank you very much, it is updated in the revised manuscript

8. In Figure 4, clarify the definitions of panels A2 and A3.

Thank you for the comment. The A2 and A3 denote abdominal segment 2 and 3, respectively. This clarification has been provided in the panel legend (Figure 4 legend line no 831 (word format)).

Referee #2 (Comments on Novelty/Model System for Author):

Novelty is medium, representing extensive functional characterization of two missense variants in the S1-S2 loop of KCNA2. Methods are extensive and are well documented. Impact is medium, focusing on two specific variants; however, impact on RAC1-ERB1/2 signaling pathways raises potential for development of targeted interventions. Use of drosophila as a model organism for study of cardiac disease/cardiomyopathy is supported by existing literature, but chosen ex vivo experiments are not well justified in the manuscript relative to in vivo imaging options, such as OCT.

Referee #2 (Remarks for Author):

This paper identifies two rare (MAF<0.01%) variants in the potassium voltage-gated channel

gene, KCNA2, in Indian and Estonian patients with DOSA, a constellation of defects including dilated cardiomyopathy (DCM), obesity, sleep apnea. Variants were identified through exome sequencing and were validated by targeted Sanger sequencing. The Indian variant was not present in the proband's unaffected mother; and the mutation status of his affected father (d. SCD) was unknown). The variant was absent from six reference datasets, including gnomAD/ExAC/1000 genomes, as well as several other regional genetic databases. The Estonian variant was present at very low MAF in gnomAD, but absent from the regional reference populations. Impacted residues for both variants were conserved and occupied similar positions within the S1-S2 extracellular loop. Functional data is provided across several models: (1) HEK293/COS7 cells overexpressing variant KCNA2 failed to correctly localize KCNA2 to the plasma membrane, resulting in failure of whole cell current generation by patch-clamp analysis. (2) Experiments in H9C2 cardiomyocytes demonstrated increased cell sizes (hypertrophy) with elevated RNA-level expression of heart failure markers and elevated levels of activated (phosphorylated) ERK1/2, rescuable by pharmacologic treatment with RAC1 inhibitors (EHT1864 and Simvastatin). (3) Patient-derived iPSC-CMs and 3D cardiac organoids showed sarcomere disorganization alongside similar hyperactivations of RAC1-ERK1/2 signaling. (4) The T184K variant was modeled in a *Drosophila* knock-in (KI) model, previously published in *Nature* in 2005 as *mns* (minisleep), a line characterized by shortened sleep cycles and reduced lifespans. No cardiac data was presented in the 2005 publication and the *mns* phenotype is noted to be recessive. KCNA2 is, however, notably located on the X chromosome in flies, in contrast to autosomal chromosome 1 in humans. In the reviewed study, KI heart tubes showed disorganized myofibers with abnormal contractility, alongside elevated abdominal fat/lipids/triglycerides and reduced sleep. These phenotypes were rescuable with treatment with RAC1 inhibitors.

Overall this is a well-written and presented manuscript with logical presentation of functional data supporting pathogenicity of the identified variants. Support for variant causation is bolstered by pharmacological rescue across several disease models alongside requisite rarity, conservation, and *in silico* predictions of protein impact for rare-disease models. Comments to address are listed below:

Major Comments:

1. It is not clear what other genes were analyzed in these families. For DCM, close to 20 other genes have definitive to strong evidence for causation by ClinGen criteria (PMID: 33947203), with several others linked to obesity-related DCM (NOX4, HTRA1, KLHL29, (PMID: 36547458). A table should be added to the supplementary materials documenting P/LP/VUS variants in these other relevant genes. Given availability of DNA, presence or absence of any segregating variants in individual P2 (Pedigree 1) should also be documented.

Thank you for the comment. The P/LP/VUS gene variants in P3 and P2 of Pedigree 1 related to DCM, obesity and sleep apnea are provided in the revised manuscript-appendix figure S1 (line no 78-80 in revised manuscript (word format)). Accordingly, no other P/LP/VUS or segregating variants were noted.

2. A major advantage of selecting *Drosophila* as a model system for the T184K variant is that the line has already been comprehensively described in the context of abnormal sleep phenotypes. Nevertheless, cardiac experiments have not been adequately justified. Limitations of the ex vivo contractility experiments should be discussed alongside why they were chosen over in vivo imaging modalities, such as optical coherence tomography. Heart tube lumen and wall cell sizes were not visualized, precluding assessment of dilated versus hypertrophic phenotypes.

Thank you for the comment. We were limited due to the unavailability of optical coherence tomography in our facility therefore we used the ex vivo contractility experiments to establish proof of concept of overall cardiac pathogenesis and abnormality.

3. Importantly, I see no mention as to how results of previous *KI* fly studies support or otherwise inform new findings reported in this manuscript. These data should be discussed alongside any considerations of dominant versus recessive inheritance of the fly and human phenotypes.

Thank you for comment. The previous fly model has been demonstrated to be associated with reduced sleep, thereby contributing to our newer understanding of *KCNA2* related sleep disorders and therapeutic options in *DOSA* patients. This has been revised in line no 157 to 159 of revised manuscript (word format). The discussion has been revised in accordance with the other above suggestion provided in line no 223-229 of revised manuscript (word format). Unfortunately, we were unable to provide a more comprehensive explanation due to space limitations.

4. The authors should address whether any other DCM-relevant signaling pathways (AKT, JNK, etc) were considered alongside ERK1/2.

We appreciate the reviewers' suggestion and concur that additional pathways may be involved alongside ERK1/2; however, we did not investigate this possibility. As detailed throughout the manuscript, we present robust evidence supporting the RAC1-ERK1/2 signaling pathway in DCM across various models, including patient-derived cardiomyocytes, patient organoids, and *Drosophila* models. Notably, in these models, multiple RAC1 inhibitors were observed to ameliorate the DCM confirming RAC1-ERK1/2 is the primary pathway in the disease pathogenesis.

Minor Comments:

1. I found the blue color of table 1c and e difficult to read, particularly in print format. I suggest removing the color entirely.

Thank you for the suggestion, we have revised it accordingly in the revised manuscript

2. Inclusion of REVEL scores in Figure 1e would be informative.

Thank you for the suggestion, we have now included REVEL scores in Figure 1F of the revised manuscript.

References:

- Botero V & Tomchik SM (2024) Unraveling neuronal and metabolic alterations in neurofibromatosis type 1. *Journal of Neurodevelopmental Disorders* 16: 49
- Cirelli C, Bushey D, Hill S, Huber R, Kreber R, Ganetzky B & Tononi G (2005) Reduced sleep in *Drosophila* Shaker mutants. *Nature* 434: 1087–1092
- Clements J, Hens K, Merugu S, Dichtl B, de Couet HG & Callaerts P (2009) Mutational analysis of the *eyeless* gene and phenotypic rescue reveal that an intact Eyeless protein is necessary for normal eye and brain development in *Drosophila*. *Developmental Biology* 334: 503–512
- Gbelcová H, Švéda M, Laubertová L, Varga I, Vítek L, Kolář M, Strnad H, Zelenka J, Böhmer D & Ruml T (2013) The effect of simvastatin on lipid droplets accumulation in human embryonic kidney cells and pancreatic cancer cells. *Lipids in Health and Disease* 12: 126
- Gross AM, Wolters PL, Dombi E, Baldwin A, Whitcomb P, Fisher MJ, Weiss B, Kim A, Bornhorst M, Shah AC, *et al* (2020) Selumetinib in Children with Inoperable Plexiform Neurofibromas. *New England Journal of Medicine* 382: 1430–1442
- van Heyningen V & Williamson KA (2002) PAX6 in sensory development. *Hum Mol Genet* 11: 1161–1167
- Mitt M, Kals M, Pärn K, Gabriel SB, Lander ES, Palotie A, Ripatti S, Morris AP, Metspalu A, Esko T, *et al* (2017) Improved imputation accuracy of rare and low-frequency variants using population-specific high-coverage WGS-based imputation reference panel. *Eur J Hum Genet* 25: 869–876

Palamidessi A, Frittoli E, Garré M, Faretta M, Mione M, Testa I, Diaspro A, Lanzetti L, Scita G & Fiore PPD (2008) Endocytic Trafficking of Rac Is Required for the Spatial Restriction of Signaling in Cell Migration. *Cell* 134: 135–147

Sundararaj KP, Samuvel DJ, Li Y, Nareika A, Slate EH, Sanders JJ, Lopes-Virella MF & Huang Y (2008) Simvastatin suppresses LPS-induced MMP-1 expression in U937 mononuclear cells by inhibiting protein isoprenylation-mediated ERK activation. *J Leukoc Biol* 84: 1120–1129

The I, Hannigan GE, Cowley GS, Reginald S, Zhong Y, Gusella JF, Hariharan IK & Bernard A (1997) Rescue of a *Drosophila* NF1 Mutant Phenotype by Protein Kinase A. *Science* 276: 791–794

Tong JJ, Schriener SE, McCleary D, Day BJ & Wallace DC (2007) Life extension through neurofibromin mitochondrial regulation and antioxidant therapy for neurofibromatosis-1 in *Drosophila melanogaster*. *Nat Genet* 39: 476–485

Xu J, Galvanetto N, Nie J, Yang Y & Torre V (2020) Rac1 Promotes Cell Motility by Controlling Cell Mechanics in Human Glioblastoma. *Cancers* 12: 1667

14th Jan 2026

Dear Dr. Dhandapany,

Thank you for the submission of your revised manuscript to EMBO Molecular Medicine. I am pleased to inform you that we will be able to accept your manuscript pending the following final amendments:

- 1) Please address referee #2 criticisms and clearly state the limitations of the study in a separate limitation section as suggested.
- 2) Figures: During our routine image integrity checks, we observed that the cell images within the Appendix file appear pixelated under analysis. This is often a result of converting original 16-bit TIFF files to RGB format for publication. While this is not inherently problematic, it can give the impression of image alteration to critical readers. To address this, please upload the Appendix file in a higher resolution. If it is not possible to reproduce the Appendix at higher resolution, we recommend instead uploading the original blot source data with your online submission or depositing the raw files on BioStudies (<https://www.ebi.ac.uk/biostudies/sourcedata/studies>) and including the archive accession number in your Data Availability section. This will enable us to confirm the integrity of the complete figure set and enhance transparency for readers.
- 3) Source Data: Please make sure that source data for all graphs and images are provided. Currently source data for Fig 4F is missing.
- 4) In the main manuscript file, please do the following:
 - Please address all comments suggested by our data editors listed below:
 - o Data availability statement:
 1. Please note that the specific URL for PRJNA1370669 dataset is not provided in the data availability statement.
 - o Figure legends:
 1. Please note that the exact p values are not provided in the legends of figures 2B, F; 3B, 4A, B, E.
 2. Please indicate the statistical test used for data analysis in the legend of figure 3D.
 3. Please note that information related to n is missing in the legend of figure 3D.
 - Please correct the order and headings of the sections in the manuscript to: Abstract / Keywords / The Paper Explained / Introduction / Results and Discussion / Methods / Data Availability / Acknowledgements / Disclosure and Competing Interests Statement / References / Figure Legends / Tables
 - Please combine Results and Discussion as this is a Report article. Please check "Author Guidelines" for more information: <https://link.springer.com/journal/44321/submission-guidelines#cms-Article-types>
 - Add callouts for Fig 3A and B.
 - Remove the section on Supplementary Materials.
 - Move Ethics declarations to Methods.
 - Please upload Reagents and Tools Table using our .doc template. More information on how to adhere to the format as well as downloadable templates (.docx) for the Reagents and Tools Table can be found in our author guidelines: <https://www.embopress.org/page/journal/17574684/authorguide#structuredmethods>
 - Please rename "Conflict of interest" to "Disclosure and competing interests statement". We updated our journal's competing interests policy in January 2022 and request authors to consider both actual and perceived competing interests. Please review the policy <https://www.embopress.org/competing-interests> and update your competing interests if necessary.
 - Author contributions: Please remove it from the manuscript and specify author contributions in our submission system. CRediT has replaced the traditional author contributions section because it offers a systematic machine-readable author contributions format that allows for more effective research assessment. You are encouraged to use the free text boxes beneath each contributing author's name to add specific details on the author's contribution. More information is available in our guide to authors: <https://www.embopress.org/page/journal/17574684/authorguide#authorshipguidelines>
 - In data availability please use the following format to report the accession number of your data:

[data type]: [full name of the resource] [accession number/identifier] ([doi or URL or identifiers.org/DATABASE:ACCESSION])

Please check "Author Guidelines" for more information.

<https://www.embopress.org/page/journal/17574684/authorguide#availabilityofpublishedmaterial>

5) Funding: Please make sure that information about all sources of funding are complete in both our submission system and in the manuscript. The Estonian Centre of Genomics/Roadmap II, funded by the Estonian Research Council (project number TT 17) is currently missing in our submission system.

6) Tables:

- Please add a title and a legend to the table of the list of primers e.g. Table 1 and move it after the figure legends, or Appendix Table S1 and place it in the Appendix. Also, add a callout for the table in the Methods section.
- Please rename Appendix Table S13 to Appendix Table S1 or S2 and update its callouts in the main manuscript.

7) Appendix: Please remove the line numbers.

8) Synopsis:

- Synopsis image: Please resize the image to 550 px-wide x 300-600 pixels high and upload it as a high-resolution .jpeg/.png file.

- Synopsis text: Please remove it from the manuscript file and upload it as a separate .doc file.
- Please check your synopsis text and image before submission with your revised manuscript. Please be aware that in the proof stage minor corrections only are allowed (e.g., typos).

9) As part of the EMBO Publications transparent editorial process (see our Editorial at <http://embomolmed.embopress.org/content/2/9/329>), EMBO Molecular Medicine will publish online a Review Process File (RPF) to accompany accepted manuscripts. This file will be published in conjunction with your paper and will include the anonymous referee reports, your point-by-point response and all pertinent correspondence relating to the manuscript. Let us know if you want to remove or not any figures from it prior to publication. Please note that the Authors checklist will be published at the end of the RPF.

10) Please provide a point-by-point letter INCLUDING my comments as well as the reviewer's reports and your detailed responses (as Word file).

I look forward to reading a new revised version of your manuscript as soon as possible.

Yours sincerely,

Zeljko Durdevic

Zeljko Durdevic
Senior Editor
EMBO Molecular Medicine

*** Instructions to submit your revised manuscript ***

When preparing your revised manuscript, please refer to our guidelines: <https://link.springer.com/journal/44321/submission-guidelines#cms-Revised-submissions>. We perform an initial quality control of all revised manuscripts before re-review; failure to include requested items will delay the evaluation of your revision.

We require:

- 1) A .docx formatted version of the manuscript text (including legends for main figures, EV figures and tables). Please make sure that the changes are highlighted to be clearly visible.
- 2) Individual production quality figure files as .eps, .tif, .jpg (one file per figure). For guidance, download the 'Figure Guide PDF': <https://media.springernature.com/original/springer-cms/rest/v1/content/27825798/data/v1>.
- 3) A .docx formatted letter INCLUDING the reviewers' reports and your detailed point-by-point responses to their comments. As part of the EMBO Press transparent editorial process, the point-by-point response is part of the Review Process File (RPF), which will be published alongside your paper.
- 4) A complete author checklist, which you can download from our author guidelines. Please insert information in the checklist that is also reflected in the manuscript. The completed author checklist will also be part of the RPF.
- 5) Please note that all corresponding authors are required to supply an ORCID ID for their name upon submission of a revised

manuscript.

6) It is mandatory to include a 'Data Availability' section after the Materials and Methods. Before submitting your revision, primary datasets produced in this study need to be deposited in an appropriate public database, and the accession numbers and database listed under 'Data Availability'. Please remember to provide a reviewer password if the datasets are not yet public.

7) For data quantification: please specify the name of the statistical test used to generate error bars and P values, the number (n) of independent experiments (specify technical or biological replicates) underlying each data point and the test used to calculate p-values in each figure legend. The figure legends should contain a basic description of n, P and the test applied. Graphs must include a description of the bars and the error bars (s.d., s.e.m.).

9) Our journal encourages inclusion of *data citations in the reference list* to directly cite datasets that were re-used and obtained from public databases. Data citations in the article text are distinct from normal bibliographical citations and should directly link to the database records from which the data can be accessed. In the main text, data citations are formatted as follows: "Data ref: Smith et al, 2001" or "Data ref: NCBI Sequence Read Archive PRJNA342805, 2017". In the Reference list, data citations must be labeled with "[DATASET]". A data reference must provide the database name, accession number/identifiers and a resolvable link to the landing page from which the data can be accessed at the end of the reference.

12) Author contributions: You will be asked to provide CRediT (Contributor Role Taxonomy) terms in the submission system. These replace a narrative author contribution section in the manuscript.

13) A Conflict of Interest statement should be provided in the main text.

14) Every published paper includes a 'Synopsis' to further enhance discoverability. Synopses are displayed on the journal webpage and are freely accessible to all readers. They include a short stand first (maximum of 300 characters, including space) as well as 2-5 one-sentences bullet points that summarizes the paper. Please write the bullet points to summarize the key NEW findings. They should be designed to be complementary to the abstract - i.e. not repeat the same text. We encourage inclusion of key acronyms and quantitative information (maximum of 30 words / bullet point). Please use the passive voice. Please attach these in a separate file or send them by email, we will incorporate them accordingly.

15) Include a Reagents and Tools Table as part of the Methods section, which can be downloaded from our author guidelines.

Photos 400-800 DPI

*Additional important information regarding figures and illustrations can be found at <https://media.springernature.com/original/springer-cms/rest/v1/content/27825798/data/v1>

***** Reviewer's comments *****

Referee #1 (Remarks for Author):

The authors have addressed all my comments satisfactorily. In addition, the author is required to provide more detailed information regarding reagents, including fruit flies, antibodies, and chemicals.

Referee #2 (Comments on Novelty/Model System for Author):

Same comments as previously stated for original submission:

Novelty is medium, representing extensive functional characterization of two missense variants in the S1-S2 loop of KCNA2. Methods are extensive and are well documented. Impact is medium, focusing on two specific variants; however, impact on RAC1-ERB1/2 signaling pathways raises potential for development of targeted interventions. Use of drosophila as a model organism for study of cardiac disease/cardiomyopathy is supported by existing literature, but chosen ex vivo experiments are not well justified in the manuscript relative to in vivo imaging options, such as OCT.

Referee #2 (Remarks for Author):

This resubmitted paper describes two rare (MAF<0.01%) variants in the potassium voltage-gated channel gene, KCNA2, in Indian and Estonian patients with DOSA. I previously reviewed this paper in its original submitted form.

Comments:

1. Overall, the authors have addressed many of my comments, including documentation of P/LP/VUS variants identified in other known DCM/obesity/sleep apnea genes (none identified). This is sufficient assuming the same variant adjudication methodology was followed for these genes as for KCNA2. It would be helpful for this to be explicitly stated in the "Exome sequencing and analysis" section of methods.

2. The authors have not yet addressed Comment#2, Reviewer#2. It is understandable that institutional or budget limitations may dictate one strategy over another, but the limitations of the ex vivo contractility experiments need to be discussed. As was previously noted, heart tube lumen and wall sizes were not reported. If ex vivo experiments are unable to confirm dilation or hypertrophy this should be stated as a primary limitation of the available method relative to in vivo modeling (OCT). The authors additionally cannot state that the KI model recapitulates a DCM phenotype when no evidence of dilation has been presented. This may be best presented in a distinct limitations paragraph added to the discussion.

3. The authors have not yet addressed Comment#4, Reviewer#2. The authors should state in their limitations section (see comment#1 above) that while their data shows robust evidence supporting RAC1-ERK1/2 signaling across diverse experimental models and that these results were rescuable with RAC1 inhibitors, impact in other potentially DCM-relevant pathways were not assessed and cannot be ruled out. RAC1 inhibitors can impact JNK/AKT signaling (PMIDs: 18826357, 25913013, 15262848,

etc). If there are compelling reasons why these pathways or the possibility of pathway crosstalk should be excluded, these should be stated.

Referee #1 (Remarks for Author):

The authors have addressed all my comments satisfactorily. In addition, the author is required to provide more detailed information regarding reagents, including fruit flies, antibodies, and chemicals.

Thank you very much for insightful comments and time. As suggested, more detailed information regarding reagents were provided in the revised manuscript.

Referee #2 (Comments on Novelty/Model System for Author):

Same comments as previously stated for original submission:

Novelty is medium, representing extensive functional characterization of two missense variants in the S1-S2 loop of KCNA2. Methods are extensive and are well documented. Impact is medium, focusing on two specific variants; however, impact on RAC1-ERB1/2 signaling pathways raises potential for development of targeted interventions. Use of drosophila as a model organism for study of cardiac disease/cardiomyopathy is supported by existing literature, but chosen ex vivo experiments are not well justified in the manuscript relative to in vivo imaging options, such as OCT.

Referee #2 (Remarks for Author):

This resubmitted paper describes two rare (MAF<0.01%) variants in the potassium voltage-gated channel gene, KCNA2, in Indian and Estonian patients with DOSA. I previously reviewed this paper in its original submitted form.

Comments:

1. Overall, the authors have addressed many of my comments, including documentation of P/LP/VUS variants identified in other known DCM/obesity/sleep apnea genes (none identified). This is sufficient assuming the same variant adjudication methodology was followed for these genes as for KCNA2. It would be helpful for this to be explicitly stated in the "Exome sequencing and analysis" section of methods.

Thank you. We have updated the same in the methods section (Line no: 301-302 (word format)).

2. The authors have not yet addressed Comment#2, Reviewer#2. It is understandable that institutional or budget limitations may dictate one strategy over another, but the limitations of the ex vivo contractility experiments need to be discussed. As was previously noted, heart tube lumen and wall sizes were not reported. If ex vivo experiments are unable to confirm dilation or hypertrophy this should be stated as a primary limitation of the available method relative to in vivo modeling (OCT). The authors additionally cannot state that the KI model recapitulates a DCM phenotype when no evidence of dilation has been presented. This may be best presented in a distinct limitations paragraph added to the discussion.

Thank you for the reviewer's suggestion. Our ex vivo experiments demonstrated that KI flies showed apparent DCM phenotypes including disorganized myofibrillar organization, reduced contraction amplitude and velocity (Figure 4B and Appendix Figure S12C). However, as reviewer suggested, we have revised the manuscript by stating it as DOSA-like phenotypes with respect to the *Drosophila* model and removing the respective recapitulates statement in the manuscript. As suggested, we have now included a limitation section regarding OCT (line no 240-244(word format)).

3. The authors have not yet addressed Comment#4, Reviewer#2. The authors should state in their limitations section (see comment#1 above) that while their data shows robust evidence supporting RAC1-ERK1/2 signaling across diverse experimental models and that these results were rescuable with RAC1 inhibitors, impact in other potentially DCM-relevant pathways were not assessed and cannot be ruled out. RAC1 inhibitors can impact JNK/AKT signaling (PMIDs: 18826357, 25913013, 15262848, etc). If there are compelling reasons why these pathways or the possibility of pathway crosstalk should be excluded, these should be stated.

Thanks for the comment. As suggested, this has been now added in the limitation section of the manuscript (line no 244-246 (word format)).

10th Feb 2026

Dear Dr. Dhandapany,

We are pleased to inform you that your manuscript is accepted for publication and is now being sent to our publisher to be included in the next available issue of EMBO Molecular Medicine.

You may qualify for financial assistance for your publication charges - either via a Springer Nature fully open access agreement or an EMBO initiative. Check your eligibility: <https://link.springer.com/journal/44321/how-to-publish-with-us>

Zeljko Durdevic
Senior Editor
EMBO Molecular Medicine

>>> Please note that it is EMBO Molecular Medicine policy for the transcript of the editorial process (containing referee reports and your response letter) to be published as an online supplement to each paper. If you do NOT want this, you will need to inform the Editorial Office via email immediately. More information is available here: <https://link.springer.com/partners/embo-press/editorial-policies#Peer%20review>